# Blackbox Attacks via Surrogate Ensemble Search

**Zikui Cai,   Chengyu Song,   Srikanth Krishnamurthy,**
**Amit Roy-Chowdhury,   M. Salman Asif**[*]
University of California Riverside

## Abstract

Blackbox adversarial attacks can be categorized into transfer- and query-based attacks. Transfer methods do not require any feedback from the victim model, but provide lower success rates compared to query-based methods. Query attacks often require a large number of queries for success. To achieve the best of both approaches, recent efforts have tried to combine them, but still require hundreds of queries to achieve high success rates (especially for targeted attacks). In this paper, we propose a novel method for Blackbox Attacks via Surrogate Ensemble Search (BASES) that can generate highly successful blackbox attacks using an extremely small number of queries. We first define a perturbation machine that generates a perturbed image by minimizing a weighted loss function over a fixed set of surrogate models. To generate an attack for a given victim model, we search over the weights in the loss function using queries generated by the perturbation machine. Since the dimension of the search space is small (same as the number of surrogate models), the search requires a small number of queries. We demonstrate that our proposed method achieves better success rate with at least $30\times$ fewer queries compared to state-of-the-art methods on different image classifiers trained with ImageNet (including VGG-19, DenseNet-121, and ResNext-50). In particular, our method requires as few as 3 queries per image (on average) to achieve more than a $90\%$ success rate for targeted attacks and 1–2 queries per image for over a $99\%$ success rate for untargeted attacks. Our method is also effective on Google Cloud Vision API and achieved a $91\%$ untargeted attack success rate with 2.9 queries per image. We also show that the perturbations generated by our proposed method are highly transferable and can be adopted for hard-label blackbox attacks. Furthermore, we argue that BASES can be used to create attacks for a variety of tasks and show its effectiveness for attacks on object detection models. Our code is available at https://github.com/CSIPlab/BASES.

## 1   Introduction

Deep neural network (DNN) models are known to be vulnerable to adversarial attacks [1–4]. Many methods have been proposed in recent years to generate adversarial attacks [2, 5–11] (or to defend against such attacks [6, 12–20]). Attack methods for blackbox models can be divided into two broad categories: transfer- and query-based methods. Transfer-based methods generate attacks for some (whitebox) surrogate models via backpropagation and test if they fool the victim models [3, 4]. They are usually agnostic to victim models as they do not require or readily use any feedback; and they often provide lower success rates compared to query-based methods. On the other hand, query-based attacks achieve high success rate but at the expense of querying the victim model several times to find perturbation directions that reduce the victim model loss [21–25]. One possible way to achieve a high success rate while keeping the number of queries small, is to combine the transfer and query attacks. While there has been impressive recent work along this direction [26, 10, 27, 11], the

---

[*]Corresponding authors: Zikui Cai (`zcai032@ucr.edu`) and M. Salman Asif (`sasif@ucr.edu`)

36th Conference on Neural Information Processing Systems (NeurIPS 2022).

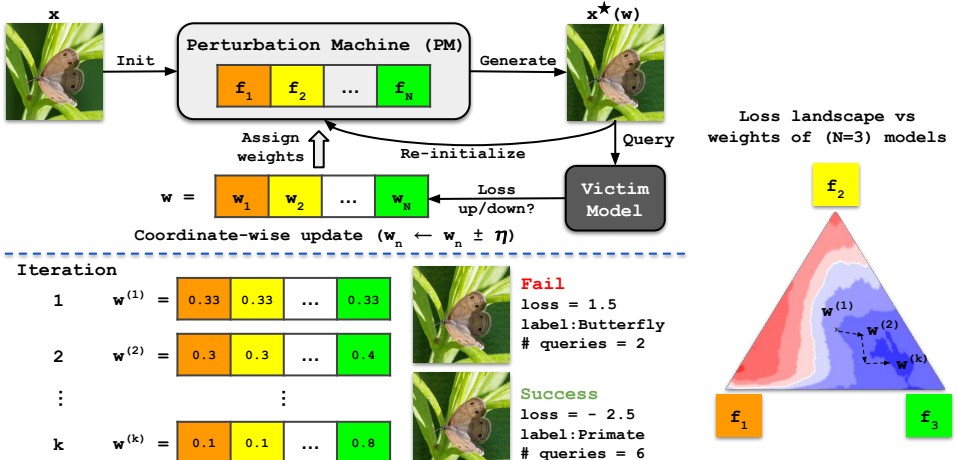

**Figure 1:** BASES for score-based attack. (**Top-left**) We define a perturbation machine (PM) using a fixed set of $N$ surrogate models, each of which is assigned a weight value as $\mathbf{w} = [w_1, \ldots, w_N]$. The PM generates a perturbed image $x^\star(\mathbf{w})$ for a given input image $x$ by minimizing the perturbation loss that is defined as a function of $\mathbf{w}$. To fool a victim model, we update one coordinate in $\mathbf{w}$ at a time while querying the victim model using $x^\star(\mathbf{w})$ generated by the PM. We can view this approach as a bi-level optimization or search procedure; the PM generates a perturbed image with the given weights $x^\star(\mathbf{w})$ in the inner level, while we update $\mathbf{w}$ in the outer level. (**Bottom-left**) We visualize weights and perturbed images for a few iterations. We stop as soon as the attack is successful (e.g. original label - 'Butterfly' is changed to target label - 'Primate' for targeted attack). (**Right**) Victim loss values for different weights along the Barycentric coordinates on the triangle. We start with equal weights (at the centroid) and traverse the space of $\mathbf{w}$ to reduce loss (concentrate on model $f_3$). Red color indicates large loss values (unsuccessful attack), and blue indicates low loss (successful attack).

state-of-the-art methods [27, 11] still require hundreds of or more queries to be successful at targeted attacks. Such attacks are infeasible for limited-access settings where a user cannot query a model that many times[28].

Given this premise, we design a new method for blackbox attacks via surrogate ensemble search (BASES), combining transfer and query ideas, to fool a given victim model with higher success rates and fewer queries compared to state-of-the-art methods. For example, our evaluation shows that BASES (on average) only requires 3 queries per image to achieve over 90% success rate for targeted attacks, which is at least $30\times$ fewer queries compared to state-of-the-art methods [10, 11]. BASES consists of two key steps that can be viewed as bilevel optimization steps. 1) A perturbation machine generates a query for the victim model based on weights assigned to the surrogate models. 2) The victim model's feedback is used to change weights of the perturbation machine to refine the query. Figure 1 depicts these steps.

We first define a perturbation machine (PM) that generates a single perturbation to fool all the (whitebox) models in the surrogate ensemble. We use a surrogate ensemble for two reasons: 1) It is known to provide better transfer attacks [29, 7]. The assumption is that if an adversarial image can fool multiple surrogate models, then it is very likely to fool a victim model as well. For the same reason, an ensemble with different and diverse surrogate models provides better attack transfer. 2) Our main interest is in searching for perturbations that can fool the given victim model. A single surrogate model provides a fixed perturbation; hence, it does not offer flexibility to search over perturbations. To facilitate search over perturbations, we define the adversarial loss for the PM as a function of weights assigned to each model in the ensemble. By changing the weights of the loss function, we can generate different perturbations and steer in a direction that fools the victim model. It is worth noting that perturbations generated by a surrogate ensemble with an arbitrary set of weights often fools all the surrogate models, but they do not guarantee success on unseen victim models; therefore, searching over the weights space for surrogate models is necessary.

Since the number of models in the surrogate ensemble is small, the search space is low dimensional and requires extremely small number of queries compared to other query-based approaches. In our method, we further simplify the search process by updating one weight element at a time, which is equivalent to coordinate descent, which has been shown to be effective in query-based attacks

[21, 23]. Since we search along one coordinate at a time instead of estimating the full gradients, the method is extremely efficient in terms of query count. In particular, our method requires two queries per coordinate update but offers success rates as good as that given by performing a full gradient update step (as shown in Section 4). Reducing the dimension of the search space while maintaining high success rate for query-based attacks is an active area of research [23, 10, 27, 11], and our proposed method pushes the boundary in this area.

We perform extensive experiments for (score-based) blackbox attacks using a variety of surrogate and blackbox victim models for both targeted and untargeted attacks. We select PyTorch Torchvision [30] as our model zoo, which contains 56 image classification models trained on ImageNet [31] that span a wide range of architectures. We demonstrate superior performance by a large margin over state-of-the-art approaches, especially for targeted attacks. Furthermore, we tested the perturbations generated by our method for attacks on hard-label classifiers. Our results show that the perturbations generated by our method are highly transferable. We present also present experiments for attacks on object detectors in the supplementary material, which demonstrate the effectiveness of our attack method for tasks beyond image classification.

The main contributions of this paper are as follows.

- We propose a novel, yet simple method, BASES, for effective and query-efficient blackbox attacks. The method adjusts weights of the surrogate ensemble by querying the victim model and achieves high fooling rate targeted attack with a very small number of queries.

- We perform extensive experiments to demonstrate that BASES outperforms state-of-the-art methods [26, 10, 27, 11] by a large margin; over 90% targeted success rate with less than 3 queries, which is at least $30\times$ fewer queries than other methods.

- We also demonstrate the effectiveness under a real-world blackbox setting by attacking Google Cloud Vision API and achieve 91% untargeted fooling rate with 2.9 queries ($3\times$ less than [10]).

- The perturbations from BASES are highly transferable and can also be used for hard-label attacks. In this challenging setting, we can achieve over 90% fooling rate for targeted and almost perfect fooling rate for untargeted attacks on a variety of models using less than 3 and 2 queries, respectively.

- We show that BASES can be used for different tasks by creating attacks for object detectors that significantly improve the fooling rate over transfer attacks.

## 2   Related work

**Ensemble-based transfer attacks.** Transferable adversarial examples that can fool one model can also fool a different model [3, 4, 29, 32]. Transfer-based untargeted attacks are considered 'easy' since the adversarial examples can disrupt feature extractors into unrelated directions (e.g., in MIM [7], the fooling rate for some models can be as high as $87.9\%$). In contrast, transfer-based targeted attacks often suffer from low fooling rates (e.g., MIM shows a transfer rate of about $20\%$ at best). To improve the transfer rate, several methods use ensemble based approach. To combine the information from different surrogate models; [29] fuses probability scores and [7] proposes combining logits. While these methods have been effective, the most natural and generic approach is to combine losses, which can be used for tasks beyond classification [33–35]. MGAA [36] iteratively selects a set of surrogate models from an ensemble and performs meta train and meta test steps to reduce the gap between whitebox and blackbox gradient directions. Simulator-Attack method [37] uses several surrogate models to train a generalized substitute model as a "Simulator" for the victim model; however, training such simulators is computationally expensive and difficult to scale to large datasets. Previous ensemble approaches typically assign equal weights for each surrogate model. In contrast, we update weights for different surrogate models based on the victim model feedback.

**Query-based attacks.** Unlike transfer-based attacks, query-based attacks do not make assumptions that surrogate models share similarity with the victim model. They can usually achieve high fooling rates even for targeted attacks (but at the expense of queries) [21, 25, 22]. The query complexity is proportional to the dimension of the search space. Queries over the entire image space can be extremely expensive [21], requiring millions of queries for targeted attack [22]. To reduce the query complexity, a number of approaches have attempted to reduce the search space dimension or leverage transferable priors or surrogate models to generate queries [23, 10, 27, 38, 39]. SimBA-DCT [23] searches over the low DCT frequencies. P-RGF [26] utilizes surrogate gradients as a transfer-based

prior, and draws random vectors from a low-dimensional subspace for gradient estimation. TREMBA [10] trains a perturbation generator and traverses over the low-dimensional latent space. ODS [27] optimizes in the logit space to diversify perturbations for the output space. GFCS [11] searches along the direction of surrogate gradients, and falls back to ODS if surrogate gradients fail. Some other methods [38, 40, 37] also reuse the query feedback to update surrogate models or 'blackbox simulator', but such a fine-tuning process provides very slight improvements. We summarize the typical search space and average number of queries for some state-of-the-art methods in Table 1. In our approach, we further shrink the search dimension to as low as the number of models in the ensemble. Since our search space is dense with adversarial perturbations, we show that a moderate-size ensemble with 20 models can generate successful targeted attacks for a variety of victim models while requiring only 3 queries (on average), which is at least 30 time fewer than that of existing methods.

## 3 Method

### 3.1 Preliminaries

We use additive perturbation [1, 2, 6] to generate a perturbed image as $x^\star = x + \delta$, where $\delta$ denotes the perturbation vector of same size as input image $x$. To ensure that the perturbation is imperceptible to humans, we usually constrain its $\ell_p$ norm to be less than a threshold, i.e., $\|\delta\|_p \leq \varepsilon$, where $p$ is usually chosen from $\{2, \infty\}$. Such adversarial attacks for a victim model $f$ can be generated by minimizing the so-called adversarial loss function $\mathcal{L}$ over $\delta$ such that the output $f(x + \delta)$ is as close to the desired (adversarial) output as possible. Specifically, the attack generator function maps the input image $x$ to an adversarial image $x^\star$ such that the output $f(x^\star)$ is either far/different from the original output $y$ for untargeted attacks, or close/identical to the desired output $y^\star$ for targeted attacks.

Let us consider a multi-class classifier $f(x) : x \mapsto z$, where $z = [z_1, \ldots, z_C]$ represents a logit vector at the last layer. The logit vector can be converted to a probability vector $p = \text{softmax}(z)$. We refer to such a classifier as a "score-based" or "soft-label" classifier. In contrast, a "hard-label" classifier provides a single label index out of a total of $C$ classes. We can derive the hard label from the soft labels as $y = \arg\max_c f(x)_c$. For untargeted attacks, the objective is to find $x^\star$ such that $\arg\max_c f(x^\star)_c \neq y$. For targeted attacks, the objective is to find $x^\star$ such that $\arg\max_c f(x^\star)_c = y^\star$, where $y^\star$ is the target label.

Many efforts on adversarial attacks use iterative variants of the fast signed gradient method (FGSM) [2] because of their simplicity and effectiveness. Notable examples include I-FGSM [5], PGD [6], and MIM [7]. We use PGD attack in our PM, which iteratively optimizes perturbations as

$$\delta^{t+1} = \Pi_\varepsilon \left( \delta^t - \lambda \, \mathbf{sign}(\nabla_\delta \mathcal{L}(x + \delta^t, y^\star)) \right), \tag{1}$$

where $\mathcal{L}$ is the loss function and $\Pi_\varepsilon$ denotes a projection operator. There are many loss functions suitable for crafting adversarial attacks. We mainly employ the following margin loss, which has been shown to be effective in C&W attacks [41]:

$$\mathcal{L}(f(x), y^\star) = \max \left( \max_{j \neq y^\star} f(x)_j - f(x)_{y^\star}, -\kappa \right), \tag{2}$$

where $\kappa$ is the margin parameter that adjusts the extent to which the example is 'adversarial.' A larger $\kappa$ corresponds to a lower optimization loss. One advantage of C&W loss function is that its sign directly indicates whether the attack is successful or not ($+ve$ value indicates failure, $-ve$ value indicates success). Cross-entropy loss is also a popular loss function to consider, which has similar performance as margin loss (comparison results are provided in the supplementary material).

### 3.2 Perturbation machine with surrogate ensemble

**Controlled query generation with PM.** We define a perturbation machine (PM) to generate queries for the victim model as shown in Figure 1. The PM accepts an image and generates a perturbation to fool all the surrogate models. Furthermore, we seek some control over the perturbations generated by the PM to steer them in a direction that fools the victim model. To achieve these goals, we construct the PM such that it minimizes a weighted adversarial loss function over the surrogate ensemble.

**Adversarial loss functions for ensembles.** Suppose our PM consists of $N$ surrogate models given as $\mathcal{F} = \{f_1, \ldots, f_N\}$, each of which is assigned a weight in $\mathbf{w} = [w_1, \ldots, w_N]$ such that $\sum_{i=1}^N w_i = 1$.

For any given image $x$ and $\mathbf{w}$, we seek to find a perturbed image $x^\star(\mathbf{w})$ that fools the surrogate ensemble. Below we discuss three possible weighted ensemble loss functions-based optimization problems for targeted attack. Loss functions for untargeted attack can be derived similarly.

$$\textbf{weighted probabilities} \qquad x^\star(\mathbf{w}) = \underset{x}{\arg\min} - \log\left(\mathbf{1}_{y^\star} \cdot \sum_{i=1}^{N} w_i \, \text{softmax}(f_i(x))\right), \qquad (3)$$

$$\textbf{weighted logits} \qquad x^\star(\mathbf{w}) = \underset{x}{\arg\min} \; \mathcal{L}(\sum_{i=1}^{N} w_i f_i(x), y^\star), \qquad (4)$$

$$\textbf{weighted loss} \qquad x^\star(\mathbf{w}) = \underset{x}{\arg\min} \; \sum_{i=1}^{N} w_i \mathcal{L}(f_i(x), y^\star). \qquad (5)$$

$y^\star$ denotes the target label and $\mathbf{1}_{y^\star}$ denotes its one-hot encoding. $\mathcal{L}$ represents some adversarial loss function (e.g., C&W loss). The first problem in (3) is the minimization of the softmax cross-entropy loss defined on the weighted combination of probability vectors from all the models in the ensemble [29]. The second problem in (4) optimizes adversarial loss over a weighted combination of logits from the models [7]. The third problem in (5) optimizes a weighted combination of adversarial losses over all models. The weighted loss formulation is the simplest and most generic ensemble approach that works not only for the classification task with logit or probability vectors, but also other tasks (e.g., object detection, segmentation) as long as the model losses can be aggregated [34]. Here, we focus on the weighted loss formulation, since it shows superior performance compared to weighted probabilities and logits formulations in our experiments (additional experiments are presented in the supplementary material).

Algorithm 1 presents a pseudocode for the PM module for a fixed set of weights. The PM accepts an image $x$ and weights $\mathbf{w}$ along with the surrogate ensemble and returns the perturbed image $x^\star = x + \delta$ after a fixed number of signed gradient descent steps (denoted as $T$) for the ensemble loss.

---

**Algorithm 1** Perturbation Machine: $\delta, x^\star(\mathbf{w}) = \mathbf{PM}(x, \mathbf{w}, \delta_{\text{init}})$

---

**Input:**
    Input $x$ and the target class $y^\star$ (for untargeted attack $y^\star \neq y$ );
    Surrogate ensemble $\mathcal{F} = \{f_1, f_2, ..., f_N\}$;
    Ensemble weights $\mathbf{w} = \{w_1, w_2, ..., w_N\}$;
    Initial perturbation $\delta_{\text{init}}$; Step size $\lambda$; Perturbation norm ($\ell_2/\ell_\infty$) and bound $\varepsilon$;
    Number of signed gradient steps $T$
**Output:** Adversarial perturbation $\delta, x^\star(\mathbf{w})$
 1: $\delta = \delta_{\text{init}}$
 2: **for** $t = 1$ to $T$ **do**
 3:     Calculate $\mathcal{L}_{\mathbf{ens}} = \sum_{i=1}^{N} w_i \mathcal{L}_i(x + \delta, y^\star)$                   ▷ *Ensemble loss*
 4:     Update $\delta \leftarrow \delta - \lambda \cdot \mathbf{sign}(\nabla_\delta \mathcal{L}_{\mathbf{ens}})$     ▷ *Gradient of ensemble via backpropagation*
 5:     Project $\delta \leftarrow \Pi_\varepsilon(\delta)$              ▷ *Project to the feasible set of $\ell_\infty$ or $\ell_2$ ball*
 6: **end for**
 7: $x^\star(\mathbf{w}) \leftarrow x + \delta$
 8: **return** $\delta, x^\star(\mathbf{w})$

---

### 3.3 Surrogate ensemble search as bilevel optimization

Let us assume that we are given a blackbox victim model, $f_{\mathbf{v}}$, that we seek to fool using a perturbed image generated by the PM (as illustrated in Figure 1). Suppose the adversarial loss for the victim model is defined as $\mathcal{L}_{\mathbf{v}}$. To generate a perturbed image that fools the victim model, we want to solve the following optimization problem:

$$\mathbf{w} = \underset{\mathbf{w}}{\arg\min} \; \mathcal{L}_{\mathbf{v}}(f_{\mathbf{v}}(x^\star(\mathbf{w})), y^\star). \qquad (6)$$

The problem in (6) is bilevel optimization that seeks to update the weight vector $\mathbf{w}$ for the PM so that the generated $x^\star(\mathbf{w})$ fools the victim model. The PM in Algorithm 1 can be viewed as a function that solves the inner optimization problem in our bilevel optimization. The outer optimization problem searches over $\mathbf{w}$ to steer the PM towards a perturbation that fools the victim model.

**BASES: Blackbox Attacks via Surrogate Ensemble Search.** Our objective is to maximize the attack success rate and minimize the number of queries on the victim model; hence, we adopt a simple yet effective iterative procedure to update the weights $\mathbf{w}$ and generate a sequence of queries. Pseudocode for our approach is shown in Algorithm 2. We initialize all entries in $\mathbf{w}$ to $1/N$ and generate the initial perturbed image $x^\star(\mathbf{w})$ for input $x$. We stop if the attack succeeds for the victim model; otherwise, we update $\mathbf{w}$ and generate a new set of perturbed images. We follow [21] and update $\mathbf{w}$ in a coordinate-wise manner, where at every outer iteration, we select $n$th index and generate two instances of $\mathbf{w}$ as $\mathbf{w}^+, \mathbf{w}^-$ by updating $w_n$ as $w_n + \eta, w_n - \eta$, where $\eta$ is a step size. We normalize the weight vectors so that the entries are non-negative and add up to 1. We generate perturbations $x^\star(\mathbf{w}^+), x^\star(\mathbf{w}^-)$ using the PM and query the victim model. We compute the victim loss (or score) for $\{\mathbf{w}^+, \mathbf{w}^-\}$ and select the weights, the perturbation vector, and the perturbed images corresponding to the smallest victim loss. We stop if the attack is successful with any query.

---

**Algorithm 2 BASES**: Blackbox Attack via Surrogate Ensemble Search

---

**Input:**
  Input $x$ and the target class $y^\star$ (for untargeted attack $y^\star \neq y$ );
  Victim model $f_\mathbf{v}$; Maximum number of queries $Q$; Learning rate $\eta$;
  Perturbation machine (PM) with surrogate ensemble
**Output:** Adversarial perturbation $\delta, x^\star$
 1: Initialize $\delta = 0$; $q = 0$; $\mathbf{w} = \{1/N, 1/N, ..., 1/N\}$
 2: Generate perturbation via PM: $\delta, x^\star(\mathbf{w}) = \mathbf{PM}(x, \mathbf{w}, \delta)$     ▷ *first query with equal weights*
 3: Query victim model: $z = f_\mathbf{v}(x + \delta)$
 4: Update query count: $q \leftarrow q + 1$
 5: **if** $\arg\max_c z_c = y^\star$ **then**
 6:     **break**                                                     ▷ *stop if attack is successful*
 7: **end if**
 8: **while** $q < Q$ **do**
 9:     Update surrogate ensemble weights as follows.         ▷ *outer level updates weights*
10:     Pick a surrogate index $n$                             ▷ *cyclic or random order*
11:     Compute $\mathbf{w}^+, \mathbf{w}^-$ by updating $w_n$ as $w_n + \eta, w_n - \eta$, respectively
12:     Generate perturbation $x^\star(\mathbf{w}^+), x^\star(\mathbf{w}^-)$ via PM     ▷ *inner level generates query*
13:     Query victim model: $f_\mathbf{v}(x^\star(\mathbf{w}^+)), f_\mathbf{v}(x^\star(\mathbf{w}^-))$     ▷ *2 queries per coordinate*
14:     Calculate victim model loss for $\{\mathbf{w}^+, \mathbf{w}^-\}$ as $\mathcal{L}_\mathbf{v}(\mathbf{w}^+), \mathcal{L}_\mathbf{v}(\mathbf{w}^-)$
15:     Select $\mathbf{w}, \delta, x^\star(\mathbf{w})$ for the weight vector with the smallest loss
16:     Increment $q$ after every query, and stop if the attack is successful for any query
17: **end while**
18: **return** $\delta$

---

## 4 Experiments

### 4.1 Experiment setup

In this section, we present experiments on attacking the image classification task. Additional experiments on attacking object detection task can be found in the supplementary material.

**Surrogate and victim models.** We present experiments for blackbox attacks mainly using pretrained image classification models from Pytorch Torchvision [30], which is a comprehensive and actively updated package for computer vision tasks. At the time of writing this paper, Torchvision offers 56 classification models trained on ImageNet dataset [31]. These models have different architectures and include the family of VGG [42], ResNet [43], SqueezeNet [44], DenseNet[45], ResNeXt [46], MobileNet [47, 48], EfficientNet [49], Reg-Net [50], Vision Transformer [51], and ConvNeXt [52]. We choose different models as the victim blackbox models for our experiments, as shown in Figures 2, 3, and 4. To construct an effective surrogate ensemble for the PM, we sample 20 models from different families: {VGG-16-BN, ResNet-18, SqueezeNet-1.1, GoogleNet, MNASNet-1.0, DenseNet-161, EfficientNet-B0, RegNet-y-400, ResNeXt-101, Convnext-Small, ResNet-50, VGG-13, DenseNet-201, Inception-v3, ShuffleNet-1.0, MobileNet-v3-Small, Wide-ResNet-50, EfficientNet-B4, RegNet-x-400, VIT-B-16}. We vary our ensemble

size $N \in \{4, 10, 20\}$ by picking the first $N$ model from the set. In most of the experiments, our method uses $N = 20$ models in the **PM**, unless otherwise specified. We also tested a different set of models pretrained on TinyImageNet dataset, the details of which are included in the supplementary material. To validate the effectiveness of our methods in a practical blackbox setting, we also tested Google Cloud Vision API.

**Comparison with other methods.** We compare our method with some of the state-of-the-art methods for score-based blackbox attacks. TREMBA [10] is a powerful attack method that searches for perturbations by changing the latent code of a generator trained using a set of surrogate models. GFCS [11] is a recently proposed surrogate-based attack method that probes the victim model using the surrogate gradient directions. We use their original code repositories [53, 54]. For completeness, we also compare with two earlier methods, ODS [27] and P-RGF [26], that leverage transferable priors, even though they have been shown to be less effective than GFCS and TREMBA. Additional details about comparison with TREMBA and GFCS are provided in the supplementary material.

**Dataset.** We mainly use 1000 ImageNet-like images from the NeurIPS-17 challenge [55, 56], which provides the ground truth label and a target label for each image. We also provide evaluation results on TinyImageNet in the supplementary material.

**Query budget.** In this paper, we move towards a limited-access setting, since for many real-life applications, legitimate users will not be able to run many queries [28]. In contrast with TREMBA and GFCS, which set the maximum query count to $10,000$ and $50,000$, respectively, we set the maximum count to be 500 and only run our method for 50 queries in the worst case. (TREMBA also uses only 500 queries for Google Cloud Vision API to cut down the cost.)

**Perturbation budget.** We evaluated our method under both $\ell_\infty$ and $\ell_2$ norm bound, with commonly used perturbation budgets of $\ell_\infty \leq 16$ and $\ell_2 \leq 255\sqrt{0.001D} = 3128$ on a 0–255 pixel intensity scale, where $D$ denotes the number of pixels in the image. For attacking Google Cloud Vision API, we reduce the norm bound to $\ell_\infty \leq 12$ to align with the setting in TREMBA. Results for $\ell_2$ norm bound are provided in the supplementary material.

**Targeted vs untargeted attacks.** All the methods achieve near perfect fooling rates for untargeted attacks in our experiments. This is because untargeted attack on image classifiers is not challenging [11], especially when the number of classes is large. Thus, we primarily report experimental results on targeted attacks in the main text and report results for untargeted attacks in the supplementary material. We use the target labels provided in the dataset [55] in the experiments discussed in the main text. We provide additional analysis on using different target label selection methods such as 'easiest' and 'hardest' according to the confidence scores in the supplementary material.

### 4.2 Score-based attacks

**Targeted attacks.** Figure 2 presents a performance comparison of five methods for targeted attacks on three blackbox victim models. Our proposed method provides the highest fooling rate with the least number of queries. P-RGF is found to be ineffective (almost $0\%$ success) for targeted attacks under low query budgets. TREMBA and GFCS are similar in performance; TREMBA shows better performance when query count is small, but GFCS matches TREMBA after nearly 100 queries. Nevertheless, our method clearly outperforms these two powerful methods by a large margin at any level of query count. We summarize the search space dimension $\mathcal{D}$ and query counts vs fooling rate of different methods under a limited (and realistic) query budget for both the targeted and untargeted attacks in Table 1. Our method is the most effective in terms of fooling rate vs number of queries (and has the smallest search dimension). *Additional results and details about fair comparison and fine tuning of TREMBA and GFCS are provided in the supplementary material.*

**Surrogate ensemble size** $(N)$**.** To evaluate the effect of surrogate ensemble size on the performance of our method, we performed targeted blackbox attacks experiment on three different victim models using three different sizes of the surrogate ensemble: $N \in \{4, 10, 20\}$. The results are presented in Figure 3 in terms of fooling success rate vs number of queries. As we increase the ensemble size, the fooling rate also increases. With $N = 20$, the targeted attack fooling rate is almost perfect within 50 queries. Specifically, for `VGG-19` with $N = 20$, we improve from $54\%$ success rate at the first query (with equal ensemble weights) to $96\%$ success rate at the end of 50 queries; this equates to $78\%$ improvement. `DenseNet-121` and `ResNext-50` can achieve $100\%$ fooling rate with $N = 20$. With `DenseNet-121`, using 10 surrogate models, we can achieve a fooling rate of $98\%$. While

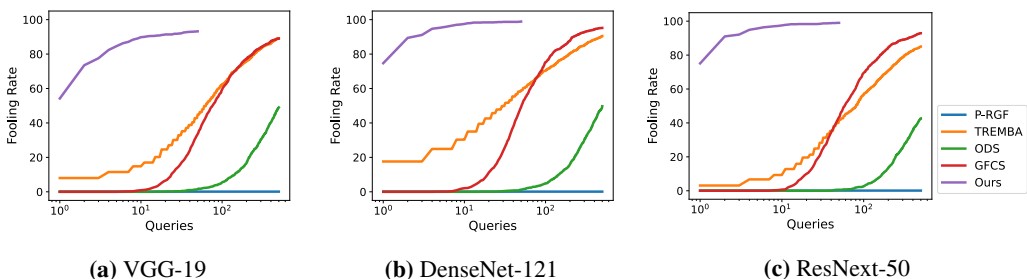

**(a)** VGG-19          **(b)** DenseNet-121          **(c)** ResNext-50

**Figure 2:** Comparison of 5 attack methods on three victim models under perturbation budget $l_\infty \leq 16$ for targeted attack. Our method achieves high success rate (over 90%) with few queries (average of 3 per image).

**Table 1:** Number of queries vs fooling rate of different methods and the search space dimension $\mathcal{D}$.

| Method | $\mathcal{D}$ | Number of queries ($\mathbf{mean} \pm \mathbf{std}$) per image and fooling rate | | | | | |
|---|---|---|---|---|---|---|---|
| | | VGG-19 | | DenseNet-121 | | ResNext-50 | |
| | | Targeted | Untargeted | Targeted | Untargeted | Targeted | Untargeted |
| P-RGF [26] | 7,500 | - | $156 \pm 113$ 
 93.5% | - | $164 \pm 112$ 
 92.9% | - | $166 \pm 116$ 
 92.5% |
| TREMBA [10] | 1,568 | $92 \pm 107$ 
 89.2% | $2.4 \pm 14$ 
 99.7% | $70 \pm 104$ 
 90.5% | $5.9 \pm 28$ 
 99.5% | $100 \pm 109$ 
 85.1% | $7.5 \pm 38$ 
 98.9% |
| ODS [27] | 1,000 | $261 \pm 125$ 
 49.0% | $38 \pm 48$ 
 99.9% | $266 \pm 123$ 
 49.7% | $52 \pm 64$ 
 99.0% | $270 \pm 116$ 
 42.7% | $54 \pm 65$ 
 98.4% |
| GFCS [11] | 1,000 | $101 \pm 95$ 
 89.1% | $14 \pm 21$ 
 100.0% | $76 \pm 75$ 
 95.2% | $16 \pm 36$ 
 99.9% | $86 \pm 87$ 
 92.9% | $15 \pm 18$ 
 99.7% |
| **Ours** | **20** | $\mathbf{3.0 \pm 5.4}$ 
 **95.9%** | $\mathbf{1.2 \pm 2.4}$ 
 **99.8%** | $\mathbf{1.8 \pm 2.7}$ 
 **99.4%** | $\mathbf{1.2 \pm 1.8}$ 
 **99.9%** | $\mathbf{1.8 \pm 2.6}$ 
 **99.7%** | $\mathbf{1.2 \pm 0.9}$ 
 **100.0%** |

using 4 models is challenging with respect to all victim models, we can see a rapid and significant improvement in fooling rates when the number of queries increases.

**Comparison of whitebox (gradient) vs blackbox (queries).** To check the effectiveness of our query-based coordinate descent approach for updating $\mathbf{w}$, we compare its performance with the alternative approach of calculating the exact gradient of victim loss under the whitebox setting. The results are presented in Figure 3 as dotted lines. We observe that our blackbox query approach provides similar results as the whitebox version, which implies the coordinate-wise update of $\mathbf{w}$ is as good as a complete gradient update.

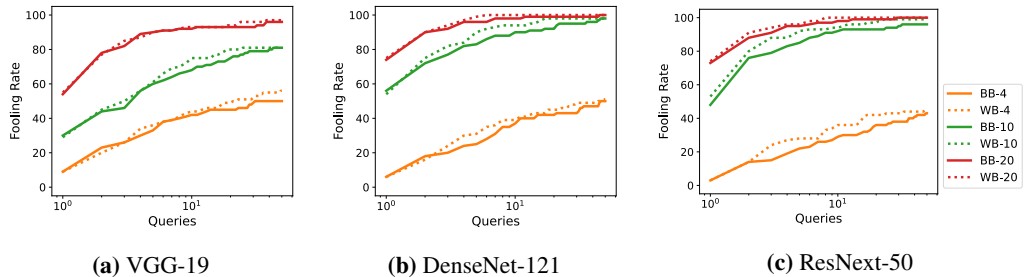

**(a)** VGG-19          **(b)** DenseNet-121          **(c)** ResNext-50

**Figure 3:** Comparison of targeted attack fooling rate with different number of ensemble models $N \in \{4, 10, 20\}$ in PM. Every experiment is performed with whitebox gradient (denoted as 'WB' with dotted lines) and blackbox score-based coordinate descent (denoted as 'BB' with solid lines). Experiment was run on 100 images.

### 4.3 Hard-label attacks

The queries generated by our PM are highly transferable and can be used to craft successful attacks for hard-label classifiers. To generate a sequence of queries for hard-label classifiers, we pick a

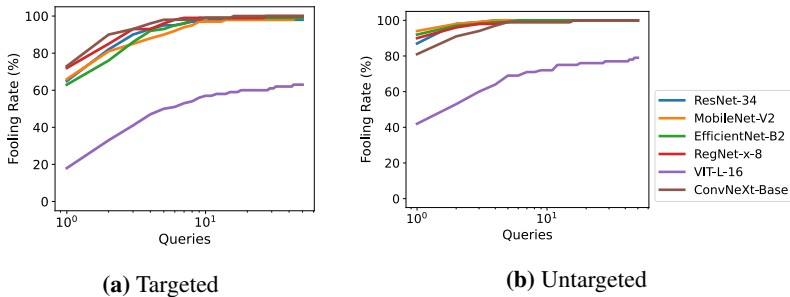

|  | **(a)** Targeted | **(b)** Untargeted |

**Figure 4:** Performance of blackbox attack on 6 hard-label classifiers. Our method generates a sequence of queries for targeted attack using `VGG-19` as a victim model while the PM has $N = 20$ models in the surrogate ensemble. Experiment performed on 100 images.

'surrogate victim' model and generate queries by updating $\mathbf{w}$ in the same manner as the score-based attacks for $Q$ iterations (without termination). We store the queries generated at every iterations in a query set $\{\delta^1, \dots, \delta^Q\}$. We test the victim hard-label blackbox model using $x + \delta$ by selecting $\delta$ from the set in a sequential order until either the attack succeeds or the queries finish.

In our experiments, we observed that this approach can achieve a high targeted attack fooling rate on a variety of models. We present the results of our experiment in Figure 4, where we report attack success rate vs query count for 6 models: {`MobileNet-V2`, `ResNet-34`, `ConvNeXt-Base`, `EfficientNet-B2`, `RegNet-x-8`, `VIT-L-16`}. We used `VGG-19` as the 'surrogate victim' model to generate the queries using the PM with 20 surrogate models. Using the saved surrogate perturbations, we can fool all models almost 100%, except for `VIT-L-16` [51] that is a vision transformer and architecturally very different from the majority of surrogate ensemble models (thus difficult to attack). Nevertheless, the fooling rate increases from $18\% \rightarrow 63\%$, which is a $250\%$ improvement.

### 4.4 Attack on commercial Google Cloud Vision API

We demonstrate the effectiveness of our approach under a practical blackbox setting by attacking the Google Cloud Vision (GCV) label detection API. GCV detects and extracts information about entities in an image, across a very broad group of categories containing general objects, locations, activities, animal species, and products. Thus, the label set is very different from that of ImageNet, and largely unknown to us. We have no knowledge about the detection models in this API either. We randomly select 100 images from the aforementioned ImageNet dataset that are correctly classified by GCV, and perform untargeted attacks against GCV using 20 surrogate models with perturbation budget of $\ell_\infty \leq 12$ to align with the setting in TREMBA [10].

For each input image, GCV returns a list of labels, which are usually the top 10 labels ranked by probability. Under the success metric of changing the top 1 label to any other label, same as in [10], our attack can achieve a fooling rate of $91\%$ with only 2.9 queries per image on average, which is much lower than 8 queries TREMBA reported for similar experiment. We present some successful examples in Figure 5. We present additional results in the supplementary material that show our attacks from classification can transfer to object detection models.

## 5 Conclusion and discussion

We propose a novel and simple approach, BASES, to effectively perform blackbox attacks in a query-efficient manner, by searching over the weight space of ensemble models. Our extensive experiments demonstrate that a wide range of models are vulnerable to our attacks at the fooling rate of over 90% with as few as 3 queries for targeted attacks. The attacks generated by our method are highly transferable and can also be used to attack hard-label classifiers. Attacks on Google Cloud Vision API further demonstrates that our attacks are generalizable beyond the surrogate and victim models in our experiments.

**Limitations.** 1) Our method needs a diverse ensemble for attacks to be successful. Even though the search space is low-dimensional, the generated queries should span a large space so that they can

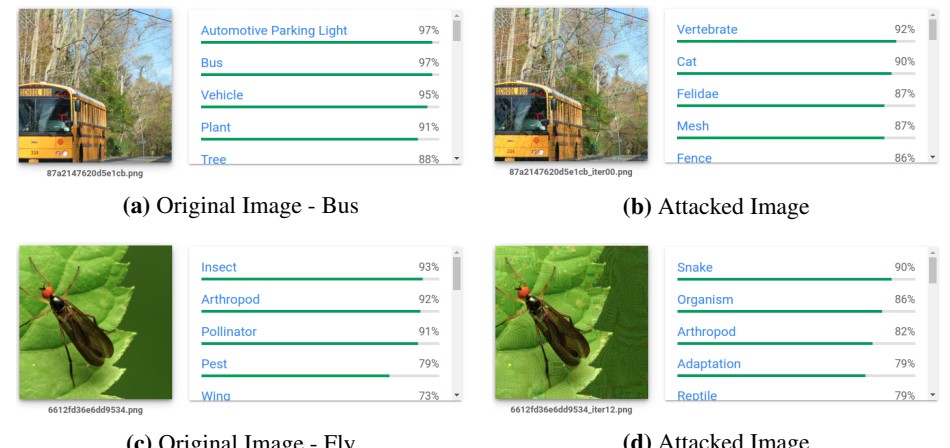

**(a)** Original Image - Bus

**(b)** Attacked Image

**(c)** Original Image - Fly

**(d)** Attacked Image

**Figure 5:** Visualization of some successful attacks on Google Cloud Vision.

fool any given victim model. This is not a major limitation for image classification task as a large number of models are available, but it can be a limitation for other tasks. 2) Our method relies on the PM to generate a perturbation query for every given set of weights. The perturbation generation over surrogate ensemble is computationally expensive, especially as the ensemble size becomes large. In our experiments, one query generation with $\{4, 10, 20\}$ surrogate models requires nearly $\{2.4s, 9.6s, 18s\}$ per image on Nvidia GeForce RTX 2080 TI. Since our method requires a small number of queries, the overall computation time of our method remains small.

**Societal impacts.** We propose an effective and query efficient approach for blackbox attacks. Such adversarial attacks can potentially be used for malicious purposes. Our work can help further explain the vulnerabilities of DNN models and reduce technological surprise. We also hope this work will motivate the community to develop more robust and reliable models, since DNNs are widely used in real-life or even safety-critical applications.

**Acknowledgments.** This material is based upon work supported by the Defense Advanced Research Projects Agency (DARPA) under agreement number HR00112090096. Approved for public release; distribution is unlimited.

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
