# Blackbox Attacks via Surrogate Ensemble Search
# Supplementary Material

## Summary

In this supplementary material, we provide additional discussion on the selection of hyper parameters, results for classification and object detection tasks, and influence of ensemble weights on the loss landscape of the victim model. Below is a summary of main sections of the supplementary material.

A. As mentioned in Section 3 of the main text, we provide experimental results justifying our selection of hyper-parameters, such as the choice of loss function for individual surrogate models, ensemble loss function, step size of PGD attack in PM, and the selection of models in the surrogate ensemble. We select the hyper-parameters that achieve the best performance for our experiments. We also discuss the effects of using different target labels.

B. As promised in Section 4 of the main text, we provide more details about the comparison with TREMBA and GFCS. We present comparisons with the Simulator Attack [37] and a hybrid attack using query-based square attack [38, 57]. We also provide additional comparisons with state-of-the-art methods for untargeted attacks and attacks under $\ell_2$ norm constraints.

C. We present experiments and results for vanishing attacks on object detectors. Our results indicate that the proposed approach is also effective for tasks beyond classification.

D. We present some examples of adversarial images generated in our experiments.

E. We analyze the effect of ensemble weights on the loss landscape of different victim models.

F. We present average top-1 classification accuracy of all the models on clean images.

## A    Analysis of hyper-parameters

### A.1    Hyper-parameters for inner optimization (PM)

Hyper-parameters can greatly impact the attack performance but can get overlooked sometimes. In our experiments, we analyzed how different hyper-parameters influence the performance of our algorithm. The experiment setup is similar to Figure 2a (in main text) using only the first 100 images to speed up experiments (because we observed similar trends using all 1000 images).

**Loss function**. Two popular candidates of loss functions (C&W, Cross Entropy) show similar performance as shown in Figure 6a. We choose C&W for the sake of convenience in determining success from its sign.

**Ensemble loss function**. Some previous papers (e.g., MIM [7]) claimed that ensemble with weighted logits (equation (4) in main text) outperforms ensemble with weighted probabilities and weighted combination of loss (equations (3) and (5) in main text). In our experiments, shown in Figure 6b, we observe that weighted combination of surrogate loss functions provide similar or even higher fooling rate compared to weighted probabilities or logits.

**PGD step size** $\lambda$. Since we are running the PGD-based attack in PM, the step size $\lambda$ can influence the attack success rate. For perturbation budget $\varepsilon = 16$ and $T = 10$ iterations, I-FGSM will use a step size of $\lambda_0 = \varepsilon/T = 1.6$ to prevent the perturbation from exceeding the $\ell_\infty$ norm bound. Since PGD projects the perturbation back to its feasible set at each iteration, we can increase the step size $\lambda$ such that more pixels saturate, which leads to a higher attack success rate. In Figure 6c, we report results using $\lambda$ as a multiple of $\lambda_0$ using multiplying factors $\{2, 3, 5, 8\}$. We choose the best step size for our experiments, which is $\lambda = 3\lambda_0$.

**Selection of surrogate models.**    As specified in the main text, our method needs a diverse ensemble for attacks to be successful.    Following the setting in TREMBA [10], we start with four surrogate models, {VGG-16-BN, ResNet-18, SqueezeNet-1.1, GoogleNet}.    To improve the diversity of our ensemble, we insert more models from different families.    We expand it to ten by    adding    {MNASNet-1.0, DenseNet-161, EfficientNet-B0, RegNet-y-400, ResNeXt-101, Convnext-Small}.    Finally,    we    add    {VGG-13, ResNet-50, DenseNet-201, Inception-v3, ShuffleNet-1.0, MobileNet-v3-Small, Wide-ResNet-50, EfficientNet-B4, RegNet-x-400, VIT-B-16} to create an ensemble with 20 models. We observe that by using 20 models in the ensemble, our method can already achieve an almost perfect targeted attack fooling rate.

### A.2    Hyper-parameters for outer optimization

**Order of surrogate models**. Since we are using a coordinate descent approach, the order of coordinates (i.e., surrogate models in our case) may influence the performance of our method. We performed different experiments

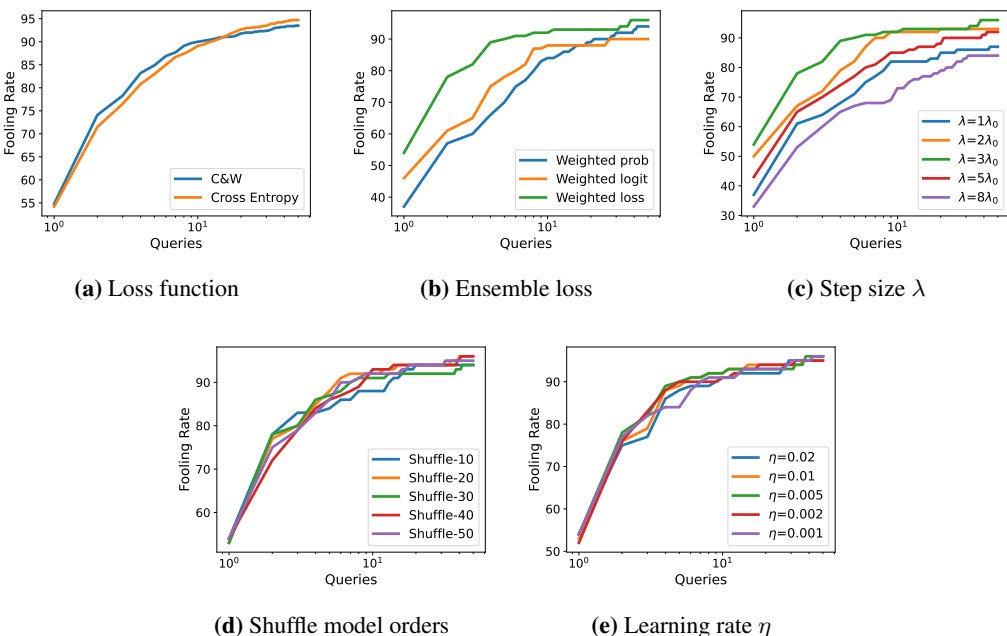

**(a)** Loss function        **(b)** Ensemble loss        **(c)** Step size $\lambda$

**(d)** Shuffle model orders        **(e)** Learning rate $\eta$

**Figure 6:** Analysis on the effect of some hyper-parameters. (a) Loss function for individual surrogate models. (b) Ensemble loss function using three types of weighted combinations. (c) Step size $\lambda$ of inner optimization in PM. (d) Order of surrogate models in PM. (e) Learning rate $\eta$ in updating ensemble weights $\mathbf{w}$ for outer optimization. All the experiment are performed for targeted attacks on victim model `VGG-19`, with ensemble size $N = 20$, and evaluated on 100 images.

by shuffling the order of the models using different random seeds (10, 20, 30, 40, 50). The results in Figure 6d suggest that our method provides identical results for different sequences of surrogate models.

**Learning rate** $\eta$. Learning rate is often an important hyper-parameter that can influence performance, and we selected our learning rate to be $^1/_{10}th$ of the average ensemble weight with 20 models (i.e., $\eta = 0.005$). We compare different learning rates in the range of $\{0.02, 0.01, 0.005, 0.002, 0.001\}$ while ensuring that all individual surrogate weights remain non-negative. The results in Figure 6e suggest that our approach is robust to variations in the learning rates.

### A.3 Selection of target labels

Different selections of target labels may result in different levels of difficulty in attacks. Here we evaluate different proposals for selecting the target labels, including the 'easiest' label (the label with the second highest original confidence score), the 'hardest' label (the label with the lowest original confidence score), and a random label. We performed an experiment to test the difficulty of three types of target labels and report our results in Table 2 below. Corresponding to Table 1 in our paper, we use `DenseNet-121` as the victim model. Here we randomly select 100 images for evaluation. We see that the original confidence scores of the NeurIPS17 target classes are already close to 0 (which means they are already challenging cases), for which our method requires an average of 1.64 queries to achieve a 100% fooling rate. The second most likely label has an average confidence of 0.08 (whereas top 1 is 0.8), and our method achieves a 100% fooling rate with an average of 1.02 queries. For the 'hardest' setting of the least confident class label, our method shows a slight drop in fooling success rate and achieves 97% success using 2.20 queries on average.

**Table 2:** Performance of BASES on different selection of target labels.

| Target classes | Avg. confidence | Fooling rate | Query counts | | | |
|---|---|---|---|---|---|---|
| | | | **mean** | **min** | **max** | **median** |
| 'Easiest' | 0.08 | 100% | 1.02 | 1 | 2 | 1 |
| NeurIPS17 | $8.92 \times 10^{-6}$ | 100% | 1.64 | 1 | 13 | 1 |
| 'Hardest' | $1.74 \times 10^{-8}$ | 97% | 2.20 | 1 | 15 | 1 |

# B  Experiments on classification

**Comparison with TREMBA.** TREMBA [10] requires one trained generator for each target class; thus, it is not feasible to test it for any arbitrary target label selected from 1000 classes in ImageNet. For a fair comparison, we attack each image using one of the 6 target labels available in trained TREMBA model $\{0, 20, 40, 60, 80, 100\}$ and average the query counts. Furthermore, TREMBA generator was trained using an ensemble of 4 surrogate models; while it is possible to train the generator with more surrogate models, training one generate per target label is expensive and non-trivial in terms of hyper-parameter tuning. Therefore, in our experiments, we used the trained generator from the paper. It is worth pointing out that our method with 4 surrogate models (as shown in Figure 3) is still better than TREMBA in the low query count regime. TREMBA can provide better success rate at the expense of increased queries.

*Why is our method better than TREMBA?* TREMBA generates patterns by optimizing over the latent code of a trained generator, which contributes to the high success rate. TREMBA generator has a large enough range that it can generate adversarial perturbations that fool a victim model. Our experimental results suggests that the space of perturbations generated by our PM (via weighted surrogate ensemble) is better (in terms of diversity and low dimensionality) than TREMBA's generator. That is the reason why we see a steep slope for the first few queries in our success vs query curve.

**Comparison with GFCS.** To perform our experiments, we used the same set of $N = 20$ surrogate models for GFCS [11] that are used in our PM. GFCS used $\ell_2$ norm constraint and did not compare with TREMBA. While our method can generate perturbations with $\ell_2$ and $\ell_\infty$ constraints, TREMBA generates perturbations with $\ell_\infty$ constraint. To perform a fair comparison, we modified GFCS code to have $\ell_\infty$ constraint and tuned the hyper-parameters to achieve the best performance. The step-size is the key parameter that we choose as $0.005$ after searching over a grid of $\{0.2, 0.02, 0.01, 0.005, 0.001, 0.0005\}$. As shown in 7, the performance reported in Figure 2 for $\ell_\infty$ attacks is on par with the performance achieved with original settings of $\ell_2$ norm constraint.

*Why is our method better than GFCS?* Our method is more query efficient because we leverage all surrogate models for each query, whereas GFCS only uses one surrogate model per query. We can see that our method has the steepest slope in Figure 7 and the highest success at the starting point.

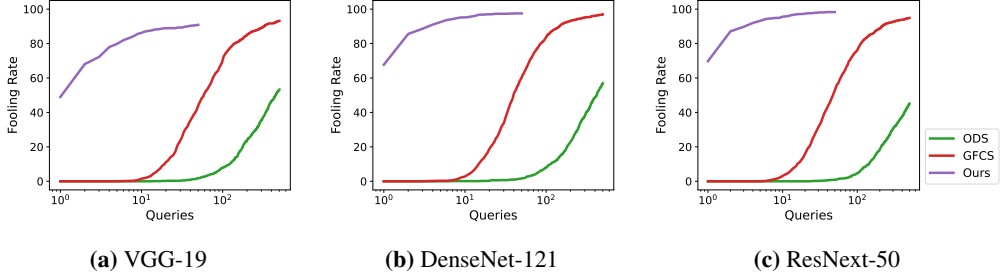

**(a)** VGG-19          **(b)** DenseNet-121          **(c)** ResNext-50

**Figure 7:** Adversarial attacks generated with $\ell_2$ constraint (equivalent to Figure 2 in main text that uses $\ell_\infty$ constraints). Comparison of our method with GFCS / ODS on three victim models under perturbation budget $\ell_2 \leq 3128$ for targeted attacks.

*Note about P-RGF.* The original implementation of P-RGF is in Tensorflow, but to unify the platform, we use the Pytorch implementation provided by GFCS [53].

**Comparison with Simulator Attack.** We use the same setting as in simulator attack [37] that tests 3 victim blackbox models {`DenseNet-121`, `ResNeXt-101 (32×4d)`, `ResNeXt-101 (64×4d)`} and uses 16 surrogate models {`VGG-11/13/16/19`, `VGG-11/13/16/19 (BN)`, `ResNet-18/34/50/101/152`, `DenseNet-161/169/201`}. All of these models are trained on TinyImageNet [58] dataset and we obtain the pretrained weights from [37]. We randomly select 1000 tinyImageNet images and use incremental target label selection for targeted attacks. Target label $y_{adv} = (y + 1) \mod C$, where $y$ is the original label and total number of classes is $C = 200$. Perturbation budget for targeted attack is $\ell_2 \leq 4.6 \times 255 = 1173$, and for untargeted attack $\ell_\infty \leq 8$. As shown in Table 3, we achieve perfect fooling rates with less than two queries on average for both targeted and untargeted attacks. Specifically, for `ResNeXt-101 (32×4d)`, we achieve $100\%$ targeted fooling rate with an average query count of 2.0, (min = 1, max = 26, median = 1). In contrast, simulator attack achieves $84.9\%$ fooling rate using 2558 queries, which is $1279\times$ more expensive than ours. For untargeted attack, the trend is similar that our method is $811$–$1445\times$ more query efficient than simulator attack.

**Comparison with combining transfer and query-based attacks.** Hybrid attack in [38] is one of the earliest works that combines transfer and query-based attacks. It uses surrogate models to generate the initial query, which is later updated using feedback from the blackbox victim model via pure query-based methods. To verify that our proposed method is advantageous, we use the perturbations generated by our ensemble models with

**Table 3:** Number of queries vs fooling rate of different methods on TinyImageNet dataset.

| Method | Number of queries (**mean/median**) per image and fooling rate | | | | | |
| | DenseNet-121 | | ResNeXt-101 (32×4d) | | ResNeXt-101 (64×4d) | |
| | Targeted | Untargeted | Targeted | Untargeted | Targeted | Untargeted |
|---|---|---|---|---|---|---|
| NES [25] | 4625 / 4337 88.5% | 1306 / 510 74.3% | 4959 / 4703 88.0% | 2104 / 765 45.3% | 4758 / 4440 88.2% | 2078 / 816 45.5% |
| Meta [59] | 5420 / 5506 24.2% | 3789 / 3202 71.1% | 5440 / 5249 21.0% | 4101 / 3712 33.8% | 5661 / 5250 18.2% | 4012 / 3649 36.0% |
| Bandits [60] | 2724 / 1860 85.1% | 964 / 520 99.2% | 3550 / 2700 72.2% | 1737 / 954 94.1% | 3542 / 2854 72.4% | 1662 / 1014 95.3% |
| Simulator [37] | 1959 / 1399 89.8% | 811 / 431 99.4% | 2558 / 1966 84.9% | 1380 / 850 96.8% | 2488 / 1982 83.9% | 1445 / 878 97.9% |
| **Ours** | **1.5 / 1 100.0%** | **1.0 / 1 100.0%** | **2.0 / 1 100.0%** | **1.0 / 1 100.0%** | **2.0 / 1 100.0%** | **1.0 / 1 100.0%** |

**Table 4:** Number of queries vs fooling rate for hybrid methods that combine transfer and query-based attacks.

| Models | Fooling rate and number of queries (**mean ± std**) per image | |
| | Combine [38] and [57] | **Ours** |
|---|---|---|
| VGG-19 | 64.7% ; 34.1 ± 99.2 | **95.9% ; 3.0 ± 5.4** |
| DenseNet-121 | 83.8% ; 24.5 ± 81.4 | **99.4% ; 1.8 ± 2.7** |
| ResNext-50 | 84.3% ; 24.7 ± 81.4 | **99.7% ; 1.8 ± 2.6** |

equal weights as the initial query, and for every failed query we deploy a powerful pure query-based method square attack [57]. Following the same setting as in our Table 1 and Figure 2, we perform targeted attack on DenseNet-121 with a perturbation budget of $\ell_\infty \leq 16$. The transfer rate of initial perturbed images is $75.5\%$. We attack the remaining $24.5\%$ failed perturbed images using square attack by allowing a maximum query count of $500$ (same setting as other baseline methods). On this subset of images, we observed a $33.9\%$ fooling rate with query count (**mean ± std**): $238.2 \pm 127.4$. Overall, including the images that can initially transfer, the combination of [38] and [57] achieves a fooling rate of $83.8\%$, with query count (**mean ± std**): $24.5 \pm 81.4$. In comparison, our method achieves a 99.4% fooling rate with a query count of $1.8 \pm 2.7$. Similar trends appear for other victim models, as shown in Table 4. Our main takeaway is that even though the surrogate ensemble provides highly transferable perturbation or perturbations that can be used as initialization for query-based optimization methods. The query-based methods lose their advantage by querying over a high dimensional image space. Our method searches over the weights of the ensemble loss, which is very low dimension and provides query efficiency.

**Untargeted attacks.** Un-targeted attacks are 'easy' [11] in image classification, especially when the number of classes is large (e.g., in ImageNet that has 1000 categories). We show that our method can readily achieve a fooling rate over 99% with only 1–2 queries (on average), as depicted in Figure 8 below and Table 1 in the main text. The initial perturbations from the PM (with all ensemble weights set to $1/N$) can already achieve a fooling rate of over $94\%$, close to that of TREMBA. Other methods require tens or hundreds of queries to achieve near-perfect success rate.

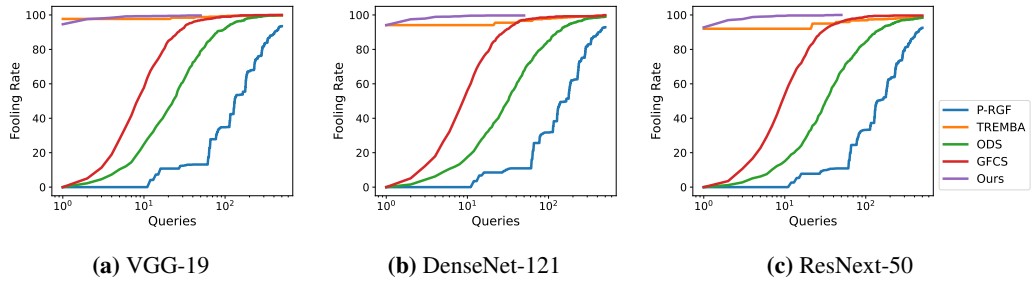

(a) VGG-19          (b) DenseNet-121          (c) ResNext-50

**Figure 8:** Untargeted attacks (version of Figure 2 in the main text). Comparison of 5 attack methods on three victim models under perturbation budget $l_\infty \leq 16$ for untargeted attack. All methods can achieve near perfect success rate within 500 queries.

# C  Experiments on object detection

To demonstrate the generalizability of BASES beyond classification tasks, we also performed experiments for vanishing attacks on object detectors. The results indicate that our proposed method can be easily adopted for other tasks.

## C.1  Experiment setup

**Surrogate and victim models.** We evaluate BASES using object detectors from MMDetection [61, 62], which provides a diverse set of models form over fifty model families, including `Faster R-CNN [63]`, `YOLOv3 [64]`, `RetinaNet [65]`, `FreeAnchor [66]`, `RepPoints [67]`, `CenterNet [68]`, `DETR [69]`, and `Deformable DETR [70]`. We choose different models {`RetinaNet, RepPoints, Deformable DETR`} as victim blackbox models, as shown in Figure 9. For surrogate models in the PM, we select some popular models {`Faster R-CNN, YOLOv3, FreeAnchor, DETR, CenterNet`} and vary our ensemble size $N \in \{2, 3, 4, 5\}$ by choosing the first $N$ models from the set.

**Dataset, attacks, query, and perturbation budgets.** All models are trained on COCO 2017 train dataset [71]. We randomly sample 100 images of stop sign from COCO 2014 validation dataset to perform blackbox vanishing attacks. The attack is considered successful if the victim model fails to detect the stop sign in the adversarial image. The constraints on the query budget $Q \leq 50$ and perturbation budget $\ell_\infty \leq 16$ are the same as the classification setting.

**Loss functions and ensemble loss.** For individual surrogate models, we use the original loss function used for their training. We defined the ensemble loss as a weighted combination of loss over all the surrogate models. The confidence score of stop sign detected by the victim model is used as a feedback from the victim model.

## C.2  Attacks on object detection

The results of attacking object detectors are shown in Figure 9 and Table 5. We observe that our attack method is effective and query efficient in attacking object detectors. In particular, for RetinaNet, a simple transfer attack (first iteration) has 27% fooling rate with $N = 2$ surrogate models. The fooling rate improve from $27\% \rightarrow 81\%$ with a small number of queries, which is a $300\%$ improvement. Our attack gets stronger as the number of surrogate models increases. When $N = 5$, we can get almost perfect ($\geq 99\%$) fooling rate for all victim models with less than 3 queries on average.

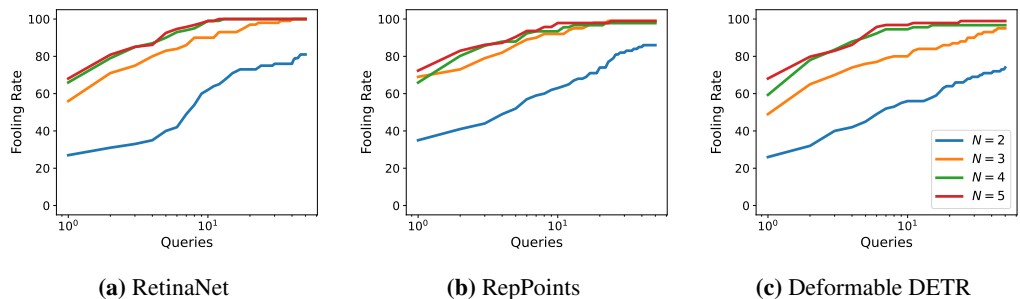

|            (a) RetinaNet            |            (b) RepPoints            |       (c) Deformable DETR       |

**Figure 9:** Fooling rates for vanishing attacks on three victim object detectors using different number ($N \in \{2, 3, 4, 5\}$) of surrogate models in PM.

**Table 5:** Number of queries per image and fooling rate of attacks on three victim models using different number $N$ of surrogate models in PM.

| $N$ | Fooling rate and number of queries ($\mathbf{mean} \pm \mathbf{std}$) per image | | |
|---|---|---|---|
|  | RetinaNet | RepPoints | Deformable DETR |
| 2 | 81% ; $8.5 \pm 11$ | 86% ; $8.0 \pm 9.9$ | 74% ; $8.5 \pm 11$ |
| 3 | 100% ; $3.9 \pm 6.5$ | 99% ; $2.8 \pm 4.1$ | 95% ; $5.4 \pm 9.3$ |
| 4 | 100% ; $2.2 \pm 2.4$ | 98% ; $2.2 \pm 3.1$ | 97% ; $2.1 \pm 2.2$ |
| 5 | 100% ; $2.0 \pm 2.1$ | 99% ; $2.1 \pm 3.0$ | 99% ; $2.1 \pm 2.9$ |

## C.3 Attacks on Google Cloud Vision API

We also observe that the attacks generated by our method can also fool object detection models, as shown in Figure 10.

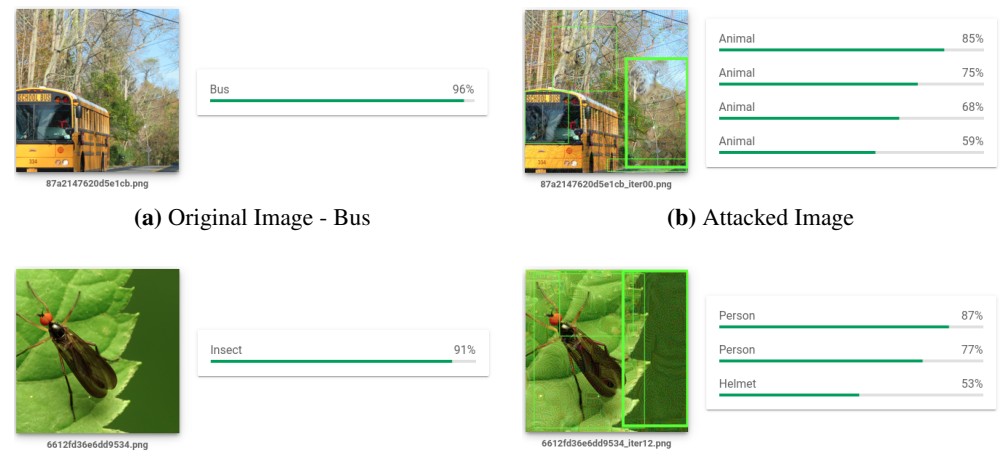

(a) Original Image - Bus

(b) Attacked Image

(c) Original Image - Fly

(d) Attacked Image

**Figure 10:** Attacks generated by our PM can fool object detection models. Visualization of some successful attacks on Google Cloud Vision object detection API. (Compare to Figure 5 in main text.)

# D  Visualization of adversarial examples

**Classifiers**. We present some examples of adversarial images generated by different methods for targeted attack on VGG-19 classifier in Figure 11. We observe that even with the same perturbation budget, $\ell_\infty \leq 16$, perturbation from our method is less visible than TREMBA, and is comparable with the ones from ODS and GFCS. TREMBA perturbs all images to 'Tench' and has a very structured semantic pattern that becomes visible. ODS, GFCS, and our method perturb 'Butterfly' to 'Dog', 'Coot' to 'Jacamar', and 'Parrot' to 'Fountain'.

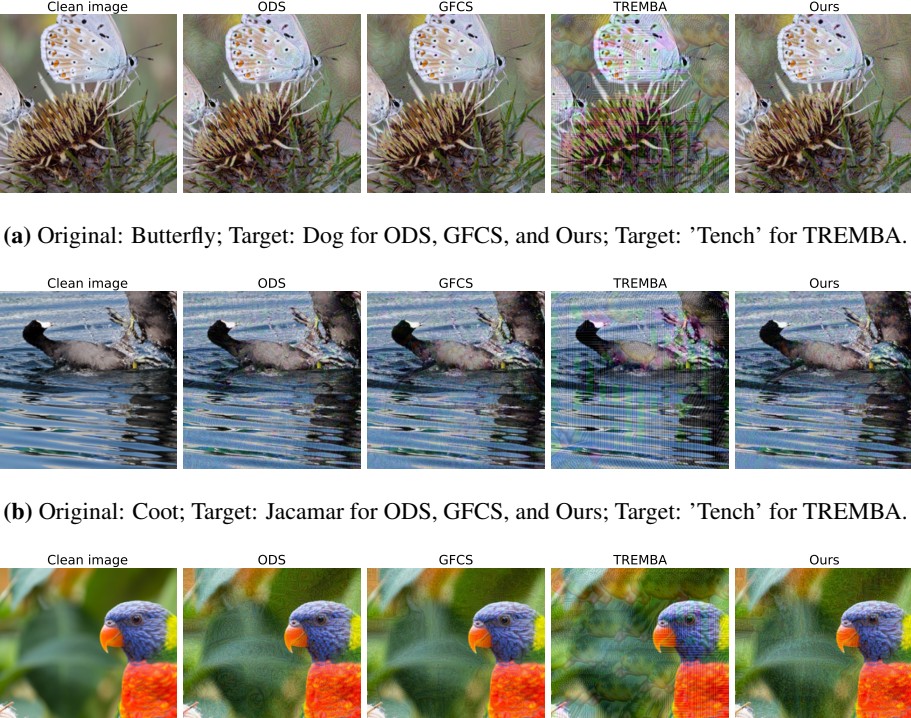

**(a)** Original: Butterfly; Target: Dog for ODS, GFCS, and Ours; Target: 'Tench' for TREMBA.

**(b)** Original: Coot; Target: Jacamar for ODS, GFCS, and Ours; Target: 'Tench' for TREMBA.

**(c)** Original: Parrot; Target: Fountain for ODS, GFCS, and Ours; Target: 'Tench' for TREMBA.

**Figure 11:** Visualization of adversarial images generated by different methods for targeted attack. (Corresponds to experiments in Figure 2 in main text.)

**Detectors**. We visualize some example images of attacking different object detectors in Figure 12. Our method effectively vanishes stop sign in the scene.

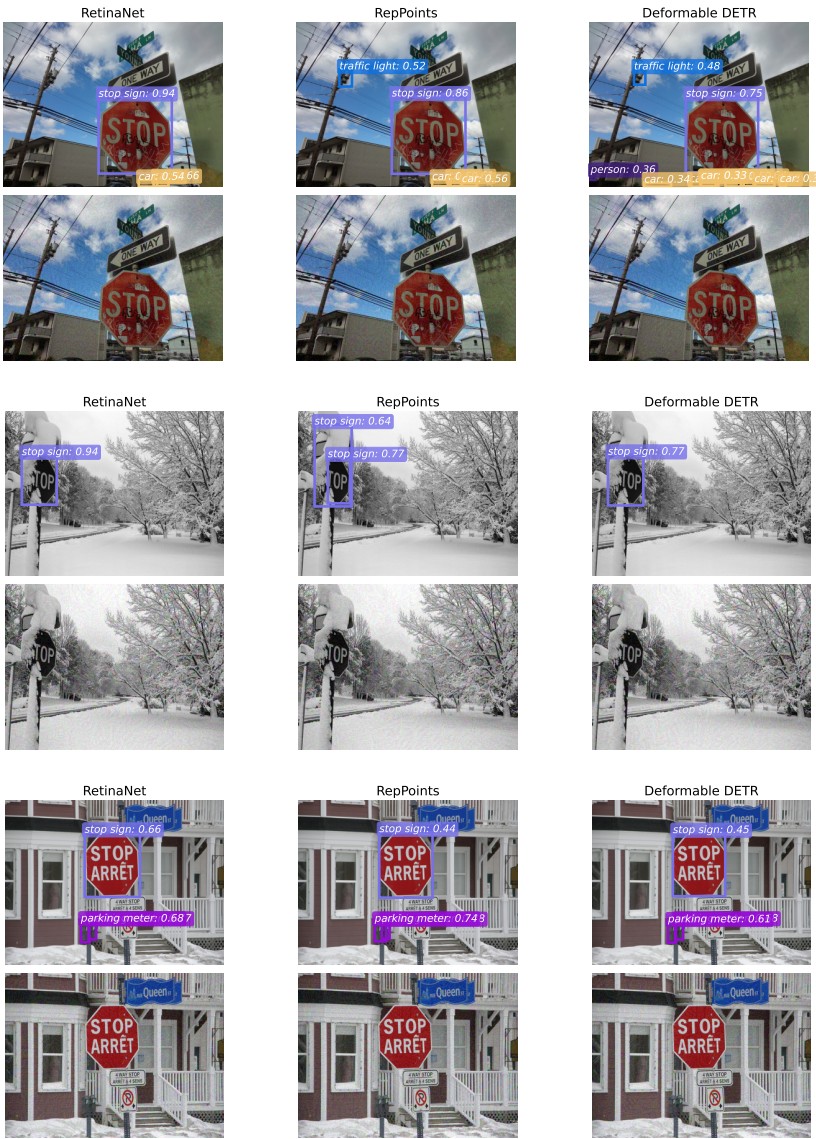

**Figure 12:** Visualization of adversarial images generated by different methods for vanishing attacks on 'stop sign'. Top row is detection on clean images and bottom row is detection on adversarial images. (Corresponds to results in Figure 9 with $N = 5$.)

# E    Loss landscape vs ensemble weights

*Why does ensemble weights-based query update work?* We visualize the loss landscape of some victim models with respect to ensemble weights of three surrogate models in the PM. The plots in Figure 13 illustrate the loss, where the vertices of each triangle represent the surrogates models in the PM used for attacking a victim model on one image (as shown in sub-caption). The location of each point inside the triangle corresponds to the weight vector $\mathbf{w}$ (in terms of Barycentric coordinates). For instance, the centroid (marked by $\times$) has the barycentric coordinate $\mathbf{w} = [1/3, 1/3, 1/3]$, which implies the losses for all the surrogate models in the ensemble are weighted equally. More weight is given to a model if the weight vector moves closer to the vertex of that model. The color of each point inside the triangle represents the victim loss value for the corresponding $\mathbf{w}$. The attack is more successful when the loss value is low (indicated by blue color) and less successful when the loss value is high (indicated by red color). We created this figure using VGG-16, ResNet-18, and SqueezeNet as Model 1,2, and 3, respectively. The main takeaway is that, in many cases, an arbitrary weight vector does not provide successful perturbation for a given victim model; therefore, we need to adjust the weights to generate successful attacks.

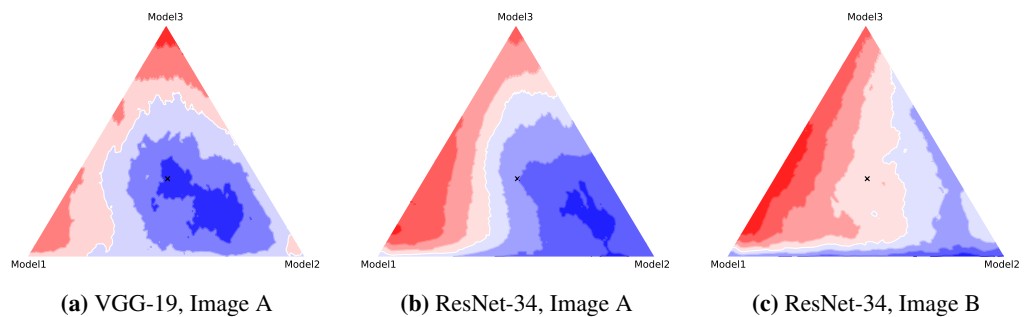

**(a)** VGG-19, Image A        **(b)** ResNet-34, Image A        **(c)** ResNet-34, Image B

**Figure 13:** Illustration of the effect of weights of ensemble models on the attack loss for a victim model. Red color indicates large loss values (unsuccessful attack), and blue indicates small loss (successful attack).

# F    Classification performance on clean images

To ensure that all the models provide reasonably correct classification results on clean images, we calculate the classification accuracy of all ImageNet models on the 1000 test images. Our calculation shows that they have a Top-1 accuracy of ($\mathbf{mean} \pm \mathbf{std}$): $89.1\% \pm 6.5\%$. Among all the models tested, `Convnext-Smal` achieves the highest accuracy at $96.8\%$, and `SqueezeNet-1.1` gets the lowest at $68.8\%$.

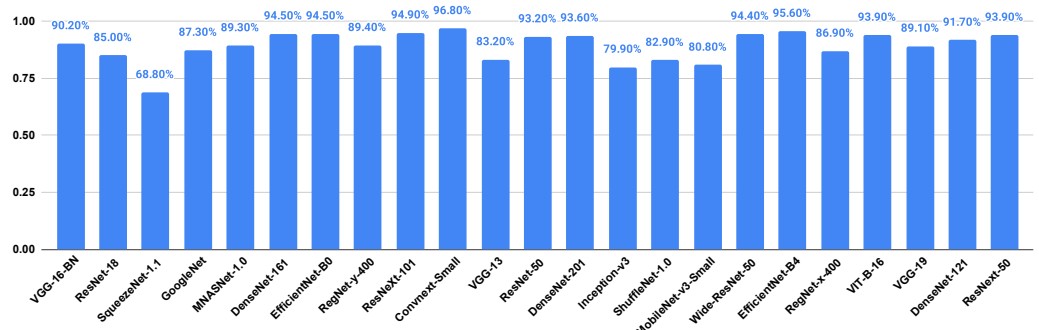

**Figure 14:** Top 1 classification accuracies of different ImageNet models used in our experiments.