# OpenReview forum: "Blackbox Attacks via Surrogate Ensemble Search"
_NeurIPS.cc/2022/Conference — NeurIPS 2022 Accept_

### Official Review · Reviewer_wjHs · 2022-07-03

**Rating:** 6
**Confidence:** 4
**Soundness:** 3 good
**Presentation:** 3 good
**Contribution:** 3 good

**Summary:**

This paper proposes a new method in generating efficient adversarial examples by a set of surrogate models. They apply PGD over the losses computed by these surrogates and a group of weights to adjust the effect of different models. In the experiments, they can find the adversarial example for each input within only a few queries, which is 30x faster than previous methods.


**Questions:**

Here are some suggestions which may help improve this paper.

1. In equation (6), the author actually uses the target label y*, which is different from previous expression trying to maximize the loss. It is suggesting that the definition of y* is re-emphasized here for clarification. Similar problem can be found in the expression of PGD optimization.

2. The main idea or the advantage of proposed method is to leverage a variety of existing models to generate adversarial examples that could fool all the surrogate models. Such examples also intend to fool the targeted victim models in large probability. The trick of the method is to transfer the consumed queries in black-box attack to white-box attack performed by a variety of surrogate models. Thus, the number of T consumed by surrogate models should be well-studied and discussed. In experiments, for more comprehensive results, it is suggesting that an appendix column to show the number of T is added in the table to make it clearer or fairer.

3. why use coordinate gradient descent for updating w instead of applying SGD over all elements in w?

4. In the appendix, the authors also study the transferability of their generated adversarial examples in the task of object detection, which is good and convincing. I got another concern that, together the black-box classification task in the first phase, is data used for training surrogates the same as the data used in attack? I mean, for example, can the surrogate be trained in the CIFAR, and the attack is performed in the ImageNet. Can we assume the attacked data is unknown in advance and different from the data used in training surrogate?

**Limitations:**

See above.


**Strengths And Weaknesses:**

The paper is easy to understand and the structure is clear. Overall, the presentation is also good.
The idea of this paper makes sense and is straightforward.

The novelty of this paper or some analysis seems a little limited and not enough, which i mean, brings limited inspiration for our future studies. To solve this, a comprehensive theoretical analysis or more experimental analysis maybe needed.

---

> ### Author Response · Authors · 2022-08-02
> **Initial response to Reviewer wjHs**
>
> Thank you for your thoughtful comments. We are glad you liked the presentation and found our paper clear and easy to understand.
>
> **(Novelty and analysis.)** We proposed and demonstrated with extensive experiments that potent adversarial examples can be generated for targeted and untargeted attacks by updating the weights of the ensemble loss. The simplicity of our approach is a feature and showcases the ease with which successful attacks can be generated. Our method requires a small number of queries (30x less than existing methods) because our search space is of very low dimension. We believe this is a significant contribution for one paper.
>
> In terms of inspiration for future studies, our method not only outperforms existing blackbox attacks on classification tasks, but can also generate attacks for other tasks (such as object detection, segmentation, etc.). How does our ensemble-based method compare against hard-label attacks? We have provided preliminary results on these ideas, but they are important directions that require further research.
>
> Our method also raises several questions for future theoretical studies. For instance, what is the space of perturbations that can be generated by an ensemble as a function of the loss weights? how can we quantify and enhance the diversity of ensembles (also related to Reviewer vphn)?
>
>
> **Reply to questions:**
>
> 1. **(Re-emphasize definition.)** Thank you for the suggestion, we will modify the text accordingly in the revision even though we have mentioned the notation of y* multiple times before eq (6), e.g. lines 129, 136, 159, algorithm1 input definition.
>
> 1. **(Value of T)** In our experiments, T denotes the number of iterations that our perturbation machine uses to minimize the ensemble loss function in Algorithm 1. We fixed T=10 in all of our experiments, as mentioned on line 527 (we will include this in the main text). Each iteration involves calculating gradients of every surrogate model, which are then used to update a single adversarial perturbation. We do not view this as “transfering the consumed queries in black-box attack to white-box attack performed by a variety of surrogate models”, because whitebox attacks on surrogate models calculate the exact gradients and update a different loss than the one optimized for the victim blackbox model. We view this as a bilevel optimization problem where the blackbox model minimizes loss in Eq. 6 over w, and Algorithm 1 provides the perturbed image for a given w. We can view this overall process as an unrolled network, where we perform T backprops on each surrogate model per blackbox model query.
>
> 1. **(Use of coordinate gradient descent.)** We use a gradient-free method to reduce the overall number of queries. In the coordinate descent approach, we update one coordinate of the weight vector (w) and generate one query for the blackbox. To estimate a full gradient over w, we would require 2N queries but that is not efficient for our method. N denotes the length of w (i.e., the number of surrogate models).
>
> 1. **(Changes in datasets and labels.)** This question has two distinct parts.
> One is different datasets; for instance, same distribution and labels but data split into different parts so that surrogate and victim models have little to no overlap.
> Second is about different/unknown label sets, which raises additional questions about how we can generate specific targeted attacks.
>
>     We used the pretrained models from Torchvision, all of which are trained on the ImageNet dataset. Using the same training data for surrogate and blackbox models is the common setting in previous works ([26,10,27,11,1*, 2*,3*,4*]).
>
>     We also performed experiments on Google Cloud Vision (GCV), which suggest that our method can perform well even if training data and labels for surrogate and victim models are different or unknown, because we have no prior knowledge about training data or model architectures for GCV.
>
>     We must add that the questions about data/label/domain mismatch are not specific to our method; they are generally applicable to all adversarial attack methods and require further research.
>
>     (Please also see our response to Reviewer vphn and Ckyu on this question)
>
> We hope our response has addressed your concerns, and you will consider increasing your ratings. If we can provide any other information, please let us know.

---

> > ### Comment · Reviewer_wjHs · 2022-08-09
> > **Thanks for the response**
> >
> > Thanks for the detailed illustration of the confusing points.
> >
> > The questions of 1,3,4 are well clarified. For question 2, I think i misunderstood the algorithm in the beginning.
> > But can I confirm two points?
> >
> > a. Since T is set to a fixed number 10, so the generated adversarial noise is not necessarily able to fool all the surrogate models when finishing all the iterations?
> >
> > b. If it is true for point (a), I just think of another setting which may need investigation. How about we generate an adversarial noise in advance which could fool all the target models, regardless of the iteration T is, then we transfer once to the target black-box model, can you show or analysis the case with that in the paper?

---

> > > ### Author Response · Authors · 2022-08-09
> > > **Regarding your question**
> > >
> > > Thank you for your response. We are glad to hear that your questions are clarified.
> > >
> > > Regarding your question if the generated adversarial noise was able to fool all the surrogate models. The answer is yes (almost always). In our perturbation machine design, we make sure that all the surrogate models are fooled simultaneously (see L150-151). We found T=10 iterations for the whitebox attack to be sufficient. In all our experiments, the average whitebox fooling rate of all surrogate models was 98–100%, across different surrogate sizes we tested.

---

> > > > ### Comment · Reviewer_wjHs · 2022-08-09
> > > > **Thanks for the explanation**
> > > >
> > > > Thanks for further clarification.
> > > >
> > > > All my concerns are well addressed and I like to raise my score to weak accept.

---

> > > > > ### Author Response · Authors · 2022-08-09
> > > > > **thank you**
> > > > >
> > > > > Thank you very much for increasing your rating.
> > > > >
> > > > > Can we please request you to also revise your scores on soundness, presentation, and contribution.

---

### Official Review · Reviewer_cKYU · 2022-07-05

**Rating:** 4
**Confidence:** 5
**Soundness:** 2 fair
**Presentation:** 3 good
**Contribution:** 2 fair

**Summary:**

This paper proposes a hybrid black-box attack on deep learning models that uses a combination of transfer and query-based attacks to reduce the number of queries while still maintaining a high targeted attack success rate. The main method involves creating a weighted ensemble of surrogate models, which is then used to generate local adversarial examples. These are tested against the victim model and the ensemble weights are updated using a form of coordinate descent based on the success of the locally generated adversarial examples. Experiments are carried out on non-robust models trained on the Imagenet dataset using standard Lp threat models. Both targeted and untargeted attacks are used. Real-world experiments on the Google Cloud Vision API also demonstrate the effectiveness of the method.

**Questions:**

- Line 262-263: Does the whitebox version involve taking gradients with respect to the weight vector $\mathbf{w}$? How is this gradient computed?
- Line 267-268: For the hard-label attacks using a surrogate victim model, is score based access to the surrogate model assumed? What is the link between the architecture of the surrogate and true victim models?
- Comparison with existing hard label attacks: Is there a reason there is no direct comparison to existing hard-label attacks such as the Hop-Skip-Jump attack (https://arxiv.org/abs/1904.02144)?


**Limitations:**

- While I appreciate that the authors discuss the runtime of the perturbation machine as a possible source of computational expense, I would have liked to see direct timing comparisons between the proposed attack and existing black-box attacks. This is particularly important since the per-query time for this attack is likely to be higher than attacks using simpler queries.

**Strengths And Weaknesses:**

**Strengths**
+ The design of the proposed attack which involves optimizing over the weights in an ensemble of models to generate transferable adversarial examples ensures, by construction, that the number of queries used will be limited due to the significant amount of prior information leveraged to construct the adversarial examples. This is borne out by the empirical results, which use a small number of queries even for high-dimensional Imagenet data.
+ The attack description is clear and well-written. The attack is easy to implement given a number of trained models with adversarial examples that transfer to the model under consideration.
+ The attack is shown to be effective against different architectures.
**Weaknesses**
- My main concern with the paper is that the evaluation is rather thin. The experiments are carried out on a single dataset, and only models with non-robust training are considered. Also, only a single modality (image classification) and vision API (Google Cloud) is considered. For a more thorough evaluation, I would recommend that the authors do the following:
     - Attempt black-box attacks on another modality such as object recognition or segmentation. This will also enable the use of a different dataset such as MS-COCO.
    - Explore the space of attacks on robustly trained models. Does using robust models in the ensemble help or hinder the generation of effective adversarial examples? Does the number of queries needed to attack a robust model increase as the model's robustness increases (with respect to a specified threat model)?
   - Carry out a comparative evaluation of the attack's effectiveness on different vision APIs. This is also important to determine the relation between the chosen surrogate models and the target model.
   - Provide further explanations of the link between the models chosen in the ensemble and the target model. It appears that the models chosen to create the ensemble are very close to those targeted. What happens when the models in the ensemble are trained on a different dataset, for example, or a smaller subset of the same dataset?
- The paper is missing a reference to very closely related work from Suya et al. (Hybrid Batch Attacks: Finding Black-box Adversarial Examples with Limited Queries).

---

> ### Author Response · Authors · 2022-08-02
> **Initial response to Reviwer cKYU  (Part 2/2)**
>
> **Reply to limitations:**
>
> Here we show a comparison of computational time across different methods for attacking densenet121 as the victim model, 20 surrogate models, evaluated on 100 images.
>
>
> | Method      | TREMBA | GFCS | ODS | P-RGF | Ours |
> |-------------|--------|------|-----|-------|------|
> | Time (mins) | 8      | 10   | 30  | 45    | 37   |
>
> TREMBA generates adversarial examples by a forward and backward pass over a generator, which can be performed relatively quickly. It takes a significant time to train the generator (which we do not include here), and the inclusion of new surrogate models requires retraining. On the other hand, our method does not require any training but involves computing adversarial gradients through backpropagation on the ensemble models, which is the most expensive step in our method. Per-query time in our method is larger than query over high-dimensional image space. Since our attacks are more query efficient, the overall computation is still comparable with other methods. Furthermore, in adversarial attacks with limited queries, the computational time on the whitebox side may not be as critical as the number of queries used to defeat the blackbox model.
>
> We hope our response has addressed your concerns, and you will consider increasing your ratings. If we can provide any other information, please let us know.

---

> ### Author Response · Authors · 2022-08-02
> **Initial response to Reviwer cKYU (Part 1/2)**
>
> First of all, thank you for your insightful comments. Thank you for summarizing the paper and its strengths. Our method uses fewer queries for two reasons: the first reason is, as you mentioned, our attacks are guided by perturbations generated via the fooling of a surrogate ensemble; the second reason is that we search over the weights of the surrogate loss function instead of  the high-dimensional image space.
> We discuss your comments and questions in detail below.
>
> 1.
> **(Evaluation and experiments.)**
>   1. **(Single modality.)** To clarify, we have shown results on object recognition in the supplementary material. Please see Section C where we present our results for attacks on different object detectors using images from MS COCO dataset.
>    **(Single dataset.)** ImageNet is one of the most comprehensive datasets in this area. Some previous papers may also use tiny images like MNIST and CIFAR10 for evaluation on some small models; however, they do not appear necessary given that the more comprehensive and complex ImageNet models are used. Recently, more and more papers only provide their evaluations on ImageNet (cite [26, 29, 1*, 3*]). Besides evaluations with ImageNet for classification tasks, we have also used the COCO dataset in the evaluation of object detectors.
>   1. **(Attacks on robustly trained models.)** We have performed experiments on Google Cloud Vision (GCV), and show that it is more challenging than subverting ImageNet models, and requires more queries. We expect that robustly-trained models will require more queries. We perform experiments on attacking the robustly trained model from [8*]. We follow the ensemble setting in our Table 1 but use their robust ResNet18 as the victim model, and observed an untargeted attack fooling rate of 92% with an average query of 2.
>   1. **(Different vision APIs.)** We mainly used GCV because it is widely used and acts as a strong blackbox model in our experiments.
> For the purpose of mimicking a true blackbox setting, GCV seems sufficient thus we do not consider other APIs. In other papers such as TREMBA [10], the evaluation is also performed only on GCV.
>   1. **(Different model architectures.)** We select ensemble models to be as diverse as possible, given that we do not have knowledge of the victim architecture. The model selection process in our experiments can be seen in the supplemental material Line 532.
> **(Different datasets.)** We directly used the pretrained models from Torchvision, all of which are trained on the ImageNet dataset. A thorough study on the effect of the data distribution mismatch is an interesting topic, but it would require a long  discussion and several experiments, which are beyond the scope of this paper.
> (Please also see our response to Reviwer vphn question 3.)
>
> 1. **(Missing reference.)** Thank you for pointing it out. We will include the reference and discussion in the paper. (Please see our response to Reviewer vphn for a discussion on this paper).
>
> **Reply to questions:**
>
> 1. Line 262-263: Yes, in the whitebox version we directly calculate gradients of the weight vector through backpropagation. We will clarify this.
> 1. Line 267-268: Yes, for the ‘surrogate victim’ model, we use score access. We do not assume that the true victim is relevant to the ‘surrogate victim’. Figure 4 shows that using VGG as the ‘surrogate victim’ can transfer well to 6 models (ResNet, MobileNet,  etc.). This result suggests that searching in the ensemble weight space can generate adversarial examples that are highly transferable across different models.
> 1. **(Comparison with existing hard label attacks.)** Existing hard-label attacks [6*,7*] start with an exemplar image that is already misclassified and move towards the source image. Thus they use an evaluation metric of perturbation magnitude vs queries, which is different from score based attacks. Since hard label attacks are more challenging than score based attacks, they usually take thousands of queries to get close to the source image. Because of these key differences, we do not perform a direct comparison with hard-label attack methods.
>
> [6*] Chen et al. “HopSkipJumpAttack: A Query-Efficient Decision-Based Attack”, ISSP 2020
>
> [7*] Li et al. “Nonlinear Gradient Estimation for Query Efficient Blackbox Attack”, AISTATS 2021
>
> [8*] Salman et al. “Do Adversarially Robust ImageNet Models Transfer Better?”, NeurIPS 2020

---

> ### Author Response · Authors · 2022-08-09
> **request for response**
>
> Dear Reviewer cKYU,
>
> We hope you had a chance to look at our response to your question. Please let us know if you have any additional questions or concerns that we can address at this stage.

---

> > ### Comment · Reviewer_cKYU · 2022-08-09
> > **Detailed but ultimately unconvincing rebuttal**
> >
> > I thank the authors for the extra details provided in the rebuttal. I appreciate the paper's contribution but believe it needs further work to be ready for publication.
> >
> > **Ensemble diversity**: The paper still needs a far more detailed discussion and results on the impact of ensemble diversity on attack success. The paper does not engage with the fact that when a new dataset comes along, the attacker is unlikely to have access to such a large number of good models, and may need to bootstrap from an ensemble of less effective models.
> >
> > **Other cloud APIs**: The fact that previous papers do not use other APIs does not preclude experiments with them in this paper.
> >
> > **Robust models**: A much more detailed discussion of this is needed. How many queries for targeted attacks are needed to approach the success of white-box attacks? What is the effect of adding robust models to the ensemble?

---

### Official Review · Reviewer_Emex · 2022-07-07

**Rating:** 4
**Confidence:** 3
**Soundness:** 2 fair
**Presentation:** 3 good
**Contribution:** 2 fair

**Summary:**

This paper proposes an ensemble search strategy for black-box adversarial attacks, which updates the weight of each ensemble member according to the queried feedback from target models. The empirical evaluation is done on the NeurIPS 2017 competition dataset (with 1,000 ImageNet-like images), using different model architectures. The results show that the query efficiency is largely improved.

**Questions:**

Please answer the listed weaknesses. My final rate may depend on the authors' response.

**Limitations:**

Yes

**Strengths And Weaknesses:**

Strengths:
- Clear writing and explanation of the proposed ensemble search strategy.
- The ensemble search strategy is simple but effective, leading to a significant empirical improvement.
- Commercial Google Cloud Vision API is also tested.

Weaknesses:
- My main concern is that in Table 1, the average query times of the proposed method (Ours) is around 1~2, which means the ensemble weights are only slightly updated. This makes me wonder if the effectiveness comes from the ensemble rather than the weight searching strategy.
- The architectures of the target model (e.g., VGG, DenseNet, ResNext in Figure 2) are included in the substitute ensemble, thus the ensemble search strategy is more like 'guessing the architecture of the target model'. This is not the common case in practice, because the substitute ensemble usually may not involve the architecture of the target model. More ablation studies on this aspect would be helpful.

---

> ### Author Response · Authors · 2022-08-02
> **Initial response to Reviewer Emex**
>
> Thank you very much for your insightful comments. Thank you for summarizing the paper and appreciating its strengths.
> 1.
> **(Effectiveness of weight search.)** The fooling rate of our method keeps increasing as we update the weights of the ensemble loss function after every query. This effect is best illustrated in Figure 2. For example, in Figure 2 (a), the fooling rate starts at 54% for VGG-19 using a surrogate ensemble with equal weights (this can be viewed as transfer attack success rate, but we count it as one query). The success rate gradually improves and nearly 90% of images are fooled after 10 queries. 96% images are fooled by the time we stop after a maximum of 50 queries (i.e., 78% improvement over transfer attack). The number of queries reported in Table 1 are averaged over 1000 images. We provide some additional statistics on min, max, and median queries for the three blackbox models for completeness.
>
> |  Model       | Targeted |      |      |       | Untargeted |      |      |       |
> |--------------|:--------:|:----:|:----:|:-----:|:----------:|:----:|:----:|:-----:|
> |              |   Avg.   | Min. | Max. | Median  |    Avg.    | Min. | Max. | Median  |
> | VGG-19       |    2.8   |   1  |  49  |   1   |     1.2    |   1  |  34  |   1   |
> | DenseNet-121 |    1.7   |   1  |  36  |   1   |     1.1    |   1  |  29  |   1   |
> | ResBext-50   |    1.8   |   1  |  47  |   1   |     1.2    |   1  |  44  |   1   |
>
> Our attack method provides a high transfer rate (with equal weights) in the beginning because we produce adversarial examples that fool all the surrogate models. The weights of our ensemble loss change quite a bit, and we illustrated this effect in Fig 1 and Fig 13-supplementary using real experiments over 3 surrogate models.
>
>
> 2.**(Architectures of target model in ensemble.)** To clarify, we do not use any duplicates of blackbox models in our surrogate (whitebox) ensemble, but our surrogate ensemble can contain the models from the same architectural family of the victim model. This is a common setting in numerous papers (e.g., TREMBA [10], GFCS [11], [2*]). Since we have a limited number of architecture families, it is a reasonable assumption that the blackbox model would be related to some families of our surrogate models in practice.
>
> Furthermore, we do not gain much by guessing the architecture of the target model. Even for the models from the same family, the targeted attack direct transfer rate is quite low. For example, the transfer rate from VGG13 to VGG19 is 17%, and the transfer rate from VGG16_bn to VGG19 is only 2%. Other individual models from non-VGG families have 0% transfer rate. We can provide a detailed table for transfer attack success rates across different models that exhibit low transfer attacks success rate (similar results reported in ref [29] or arXiv:1611.02770). Our main advantage comes from attacking an ensemble and searching in the weight space of ensemble loss, which is more efficient than previous approaches in defeating unknown blackbox models.
>
> Google cloud vision (GCV) is an example of a pure blackbox target and it is highly unlikely that our ensemble contains a model with the same architecture as the GCV model. Our attacks on GCV show that our method can generate potent attacks for (unseen/unrelated) blackbox models.
>
>
> [2*] Ma et al., "Simulating Unknown Target Models for Query-Efficient Black-box Attacks", CVPR 2021 https://arxiv.org/pdf/2009.00960.pdf
>
> We hope our response has addressed your concerns, and you will consider increasing your ratings. If we can provide any other information, please let us know.

---

> > ### Comment · Reviewer_Emex · 2022-08-06
> > **Thank you for the response**
> >
> > I appreciate the authors for the detailed response. As to the transfer success rates, the results reported by the authors (e.g., VGG13 to VGG19 is 17%, and the transfer rate from VGG16_bn to VGG19 is only 2%) are very low, which seem inconsistent with previous results [1*]. Could the authors further clarify on this point?
> >
> > [1*] Is Robustness the Cost of Accuracy? – A Comprehensive Study on the Robustness of 18 Deep Image Classification Models. ECCV 2018

---

> > > ### Author Response · Authors · 2022-08-06
> > > **single surrogate-based transfer attacks have low success (mismatch in settings)**
> > >
> > > Let us label Su et al., “Is Robustness the Cost of Accuracy? – A Comprehensive Study on the Robustness of 18 Deep Image Classification Models”, ECCV 2018 as [9*] to make the references consistent across different posts.
> > >
> > > There can be many possible explanations for this inconsistency.
> > > Most important one is that the attack settings in [9*] are different (easier) than our experiments in two main aspects: (1) perturbation budget and (2) evaluation metric. Figure 4 in [9*] has the closest setting to ours in terms of attack algorithm, but their perturbation budget is 5 times larger than ours; [9*]  used $\ell_\infty = 0.3$, whereas we used $\ell_\infty$ = 0.06 (16/255). [9*] reports transfer rate using the top-5 success rate, whereas we report top-1 success rate (which is more commonly used and more challenging). In terms of the VGG family, [9*] evaluated the transfer rate between VGG16 and VGG19, which is different from our earlier response of transfer rate between VGG16_bn (VGG16 with batch normalization) and VGG19. All these differences contribute to the difference in our reported numbers.
> > >
> > > Several other factors influence transfer attacks, and any comparison without considering all those factors will be incomplete and unfair. For instance, we can change step size, number of iterations, initialization to get different success rates for the same surrogate/victim pair. In our experiments, we assumed a pure blackbox scenario in which we use the same attack settings to generate perturbations that we tested on different victim models. Our main finding was that targeted attacks generated by a single surrogate model have a really small success rate on other victim models (including those with the same architecture as the surrogate model).
> > >
> > > We have uploaded an example code and table in an anonymous google drive below. Please feel free to run the script to reproduce the results for transfer attacks.
> > > Codes: https://drive.google.com/drive/folders/1ikYrhdWPqUrzcIHDdOf6ehVUonIGgQkE?usp=sharing
> > >
> > > To summarize, our experiments on single-model based transfer attacks revealed that targeted attacks generated by a single model have a really small transfer success rate. In fact, these experiments motivated us to look into ensemble-based attacks and improve them.
> > >
> > > Thank you very much for engaging in this discussion. We hope this explanation addresses your concerns. We are fairly confident about soundness and contributions of our work. We hope we have given you enough evidence to increase your individual and overall ratings.
> > >
> > > Please let us know if you have other questions.

---

> > > > ### Comment · Reviewer_Emex · 2022-08-07
> > > > **Response**
> > > >
> > > > The updated author response actually makes me more confused.
> > > >
> > > > If the transfer success rate heavily depends on so many factors (e.g., perturbation budget, bn, step size, number of iterations, initialization, just as claimed by the authors), then the experiment results shown in this paper are far from enough to verify the effectiveness of the proposed method.
> > > >
> > > > Several ablation studies should be include to make sure the proposed method is effective across different black-box settings, rather than a certain 'cherry pick' one (as claimed by the authors, 'we can change step size, number of iterations, initialization to get different success rates for the same surrogate/victim pair.')

---

> > > > > ### Author Response · Authors · 2022-08-07
> > > > > **clarification**
> > > > >
> > > > > I am sorry to hear that you are confused. Let us try to clarify.
> > > > >
> > > > > Our proposed method is **NOT** a transfer attack method. We use queries from the victim model to adjust the weights of the ensemble loss to generate better attacks.
> > > > >
> > > > > The response above was to the second question you asked on why the transfer attack success rate for a single surrogate model is so low. That question stemmed from your earlier comment that our method is 'guessing the model architecture'. We were talking about all the experiments we did with a single model-based surrogate models; which are **NOT** even part of the paper. We provided you the codes to verify that yourself.
> > > > >
> > > > > Our proposed method does not require any fine tuning of parameters. We are assuming a pure blackbox setting, where we use the same settings on surrogate ensemble side while attacking different victim models. That is the main point we were trying to convey.
> > > > >
> > > > > We respect your opinion that you may want to see more ablation studies, but we assure you we are not 'cherry picking' the results. This comment is hurtful. We request you to revisit the earlier comments and our response for a complete context.

---

> > > > > ### Author Response · Authors · 2022-08-07
> > > > > **ablation studies**
> > > > >
> > > > > One more thing to add that we have included ablation studies on step size, learning rate, and other aspects in Section A of the supplementary material.

---

> > > > ### Author Response · Authors · 2022-08-09
> > > > **follow up on the response**
> > > >
> > > > We hope you had a chance to read our last response. Can you please tell us if we can provide any additional clarification.
> > > > We really appreciate your time, discussion, and all the questions you have asked so far. Thanks!

---

### Official Review · Reviewer_vphn · 2022-07-12

**Rating:** 6
**Confidence:** 4
**Soundness:** 3 good
**Presentation:** 3 good
**Contribution:** 3 good

**Summary:**

This paper studies query efficient black-box attacks, which aims to generate black-box adversarial examples using least number of queries to the unknown victim model. The authors proposed to leverage ensemble of surrogate models to generate some candidate perturbations and then query the unknown model. The query feedback is then used to update the weights of individual surrogate models to produce better candidate perturbations for the next query. The empirical results demonstrate that the proposed approach can obtain reasonably high success with handful of queries.

**Questions:**

Most of the suggestions for improving the paper are listed in the "strengths and weaknesses" section.

**Limitations:**

Yes.

**Strengths And Weaknesses:**

The presentation of the paper is very clear and easy to follow. However, the paper lacks comprehensive review of the related works and also the adopted baselines are not convincing enough. Further, the experiments should be designed to illustrate the importance of each module in the attack process. More specifically,
1) many closely related works (especially attacks that combine transfer and query based attacks) are missing from the paper. A few samples are listed below [1], [2], [3], [4] and the authors are encouraged to conduct a more comprehensive survey on related works and include more recent baselines. For example, the listed baselines [1], [2] can be directly adopted as baselines for this paper and the combination method proposed by Suya et al. [3] might be coupled with the strong (should still be the state-of-the-art) query based attack [4] to form another strong baseline. The key is also to ensure that all baselines are using the same model ensembles. Since the nature of the proposed attack in this paper is "adaptive" transfer attacks using query feedback, it is really interesting to see if it is always advantageous over attacks that combine transfer and query based attacks directly.
[1] Yang et al., "Learning Black-Box Attackers with Transferable Priors and Query Feedback", NeurIPS 2021.
[2] Ma et a.., "Simulating Unknown Target Models for Query-Efficient Black-box Attacks", CVPR 2021
[3] Suya et al., "Hybrid Batch Attacks: Finding Black-box Adversarial Examples with Limited Queries", USENIX Security 2020.
[4] Andriushchenko et al., "Square Attack: a query-efficient black-box adversarial attack via random search", ECCV 2020.
2) It is quite surprising to me that, targeted attacks even using the baselines only cost less than 100 queries in most cases. It may be because the selected target class for misclassification is still not hard enough. A more comprehensive way of evaluation might be to include three types of target classes: 1) easiest class (second highest class in original confidence scores), 2) random class and 3) hardest class (lowest class in the original confidence scores).
3) at least the impact of ensemble diversity should be carefully measured. The current attack setup assumes the attacker has access to models that are trained on the exact same training data as the black-box model and also the ensemble set contains diverse enough architectures, which may not always hold in practice.
4) In the attack results, maybe it is also better to report the direct transfer rates when different ensembles have equal weights. With this, we can see how much improvement is obtained by updating the surrogate model weights.

---

> ### Author Response · Authors · 2022-08-02
> **Initial response to Reviewer vphn (Part 2/2)**
>
> 2.
> **(Difficulty of target labels.)** We did not select the target labels in our experiments. We followed the procedure outlined in the NeurIPS17 competition. We choose the evaluation images and target labels from this competition because it is a well recognized test kit that many other papers have used [7,8,9,1*]. Note that the field of adversarial attacks has undergone rapid advancements in recent years, mainly due to the reduction of search space dimensions using surrogate gradients. The fooling rates recent methods [10,27,11] can achieve within tens or hundreds of queries would have taken previous methods tens of thousands of queries. (See Table 3 in [3*]).
>
> Having said that, we appreciate the reviewer’s proposal for selecting the target labels. We performed an experiment to test the difficulty of three types of target labels and report our results below. Corresponding to Table 1 in our paper, we use DenseNet-121 as the victim model. We see that the original confidence scores of the NeurIPS17 target classes are already close to 0 (which means they are already challenging cases), for which our method requires an average of 1.64 queries to achieve a 100% fooling rate. The second most likely label has an average confidence of 0.08 (whereas top 1 is 0.8), and our method achieves a 100% fooling rate with an average of 1.02 queries. For the “hardest” setting of the least confident class label, our method shows a slight drop in fooling success rate and achieves 97% success using 2.20 queries on average.
>
> |              Target classes             | Avg. confidence | Fooling rate | Avg. count  | Min. count | Max. count | Median count |
> |:---------------------------------------:|:---------------:|:------------:|:-----------:|:----------:|:----------:|:------------:|
> | “Easiest” (second most confident class) |       0.08      |     100%     |     1.02    |      1     |      2     |       1      |
> |                NeurIPS17                |     8.92e-06    |     100%     |     1.64    |      1     |     13     |       1      |
> |    “Hardest” (least confident class)    |     1.74e-08    |      97%     |     2.20    |      1     |     15     |       1      |
>
>
> #### 3. We respond to this comment in three parts.
>
> **(Impact of ensemble diversity.)** We studied the impact of ensemble diversity in Figure 3, where we show that query efficiency improves by adding more models (with different architectures) in the surrogate ensemble. Our results suggest that it is beneficial to leverage all available models for achieving the highest query efficiency. It is an interesting question and a promising future direction to quantify ensemble diversity for different tasks.
>
> **(Training data mismatch.)** Using the same training data for surrogate and blackbox models is the common setting in previous works ([26,10,27,11,1*, 2*,3*,4*]). Furthermore, our experiments on attacking Google Cloud Vision (GCV) show that our method can still handle data mismatch situations because most likely, training data for GCV and our surrogate ensemble are different. A thorough study on the effect of data distribution mismatches is an interesting topic, but it would require longer discussion and several experiments, which are beyond the scope of this paper.
>
> **(Availability of diverse ensemble architectures.)** In practice, we have access to a large number of models with diverse architectures, especially for the widely deployed scenarios like classification/object detection for natural images. For example, we have a huge number of publicly available models for classification, object detection, and  segmentation. (See github repo of torchvision or mmcv https://github.com/open-mmlab/mmcv for a collection of more tasks and available models). Our approach offers strong attacks that directly benefit from a large and increasing number of available models. Since new models usually borrow good designs from previously published models, it would be difficult for a state-of-the-art blackbox model to completely avoid sharing any similarities with existing architectures. Our experiments on Google API shows that our attack performs well in a real blackbox setting without any knowledge of the victim model architecture.
>
> 4.**(Equal weights.)** Thank you for bringing up this question. We have actually discussed the direct transfer rates when using equal ensemble weights from line 252 to line 254 and in our figures (see Figures 2, 3, 4, 6, 7, 8, 9), where Queries = 1. We observe a large improvement by updating the surrogate model weights; for example, in Fig2 (a) the direct transfer rate (equal weights) is 54%, which increases to 96% after weights update (this is a 78% improvement). Different experiments show similar results.
>
> We hope our response has addressed your concerns, and you will consider increasing your ratings. If we can provide any other information, please let us know.

---

> > ### Comment · Reviewer_vphn · 2022-08-07
> > **Thanks for the additional clarification**
> >
> > I like the way the authors address my problems. Now, I am convinced that selection of target labels are not a factor here. Also, thanks for clarifying the misunderstanding on query=1 case. Since the attack success rate is quite high with limited queries, I suggest the authors may include a high-level summary in the paper and this may inform the readers better: the transfer attacks (given diverse enough local models) have the potential to achieve very high success rate, and we can unlock this potential by learning proper weight for each individual model using limited queries to the victim model. After the author rebuttal, I am now raising my score and the score can be further improved if the concerns on combined attack in part 1/2 can also be addressed.

---

> > > ### Author Response · Authors · 2022-08-08
> > > **Thank you for your valuable comments**
> > >
> > > We are grateful to you for acknowledging our work and increasing your scores. Thank you for your valuable suggestions, we will include the high-level summary as suggested. Please find the results for the combined attacks in our response above.

---

> ### Author Response · Authors · 2022-08-02
> **Initial response to Reviewer vphn (Part 1/2)**
>
> Thank you very much for your thoughtful comments. Thank you for nicely summarizing the paper and its strengths. We are happy to hear that you found the paper very clear and easy to follow. We discuss your concerns about evaluation/comparisons below.
>
> 1.
> **(Comparison with closely related work.)** Thank you for pointing out the 4 papers that we refer to as [1*,2*,3*,4*] below. We will be happy to cite and discuss them in the paper. As far as evaluation and comparisons are concerned, we believe that our experiments provide a comprehensive comparison with the current state-of-the-art methods in literature including these papers. We focused on methods that require a small number of queries (less than 500) and achieve a high fooling success rate (close to 90%). We compared our method against four strong baseline methods and showed advantages over all of them by a large margin. [1*,2*,3*,4*] are either already shown to be inferior to the baselines we use or they provide poor performance compared to our baselines in our preliminary studies. Below we discuss each of them in detail.
>
> [1*] Our experiments include comparisons with the most recent work GFCS [11] from ICLR 2022, which outperforms [1*] LeBA as shown in Table 1 and Figure 2 of [11]. Thus, it is reasonable to expect that our method will outperform [1*] as well.
>
> [2*] requires a large number of queries as it searches over the entire image (that is a high-dimensional space). As shown in Tables 6 and 7 of [2*], the method requires hundreds and thousands of queries for untargeted and targeted attacks, respectively for TinyImageNet images (64x64). This query count is orders of magnitude higher than our method. Another drawback of the approach in [2*] is that it requires the training of a simulator, which is computationally expensive (72 GPU hours for TinyImageNet images) and makes it difficult to scale to larger images like the ones in ImageNet dataset.
>
> We have performed some experiments with our method using the models from [2*]. We use the same 16 surrogate models in Table 14 and use densenet121 as the victim model; all models are pre-trained on tinyImageNet, as provided by their code repository. We reuse their code to select tinyImageNet images at random and use incremental target label selection for targeted attacks (following [2*], target class is set to y_adv = (y+1) mod C, where y is the original label and C=200). We use the Linf norm = 8 as the constraint. We achieve a 93% attack success rate with an average query count of 11.1 (min = 1, max = 50, median = 8). For untargeted attack, our method achieves 100% fooling rate with an average query count of 3.65 (min = 1, max = 26, median = 1). In contrast, [2*] achieves a similar untargeted success rate with an avg query count of 811, which is 222 times more expensive than our method. Table 7 uses L2 constraint for targeted attack, which is an easier setting as shown in Table 3 and 4. Under the easier setting, their method still requires 1959 queries on average, while our method only requires 9.6 queries, which is 176 times more query efficient than their method.
>
> [3*] is one of the earliest works on hybrid attacks that uses surrogate models to generate the initial query, which is later updated using feedback from the blackbox model. Fine tuning the surrogate models shows slight improvements (as reported in Table 6 of [3*]). Furthermore, as reported in Table 3, [3*] suffers from extremely high query costs (over 10k queries). TREMBA has detailed experiments demonstrating that directly combining transfer and query based attacks will suffer from high query counts (see [10] Figure 1 & 2). The legend labels starting with “Trans-” are the methods that initialize query using transfer attacks. Since these methods are searching in a high-dimensional image space, the fooling rate vs query curve is almost flat (i.e., need a large number of queries to achieve a high fooling rate). Whereas the slope of (fooling rate versus query) curve with our method is much steeper compared to other methods (i.e., the fooling rate rapidly increases with a small number of queries).
>
> [4*] Square attack also searches in high-dimensional image and requires hundreds of queries for untargeted attack (Table 2 in [4*]) and thousands of queries for targeted attack (Table 9 in [4*]). This method underperforms [1*] as shown in Table 1 of [1*]. It also underperforms ODS [27] as shown in Table 5 of [27], especially in the targeted attack cases. Our method outperforms ODS by a large margin; thus, we expect our method to perform better than this method.

---

> > ### Comment · Reviewer_vphn · 2022-08-07
> > **Thanks for the detailed response**
> >
> > I appreciate the efforts from the authors in addressing my concerns. A few more comments and some remaining concerns:
> > 1. I listed the four papers just as a sample list with a hope that the authors can properly position themselves in the related work by including more works (not just limited to the listed samples), especially the ones that attempt to combine the transfer and (pure) query based attacks in some forms.
> > 2. For the combination of works [3*] and [4*], I am not fully convinced that TREMBA can directly outperform their combination. The main reasoning is, TREMBA is still constrained on the gradient estimation attack form (as in the well-known NES attack), while (at least) for $\ell_\infty$ attacks, the random search methods seem to outperform the gradient estimation based attacks significantly. Also, directly mentioning extremely high query cost for [3*] might be unfair, as the experimental settings are different (i.e., local ensembles, selected target class). It will be ideal (and I will be fully convinced) if the authors can quickly run an experiment that combines [3*] and [4*], where the local ensembles are exactly the same as the ones used in this submission, and the for the failed transfers, the Square-attack is deployed to find successful adversarial examples.
> > 3. Direct comparison of a pure query-based attack and some form of combined attack cannot be used as a convincing argument here (e.g., comparison of LeBA, ODS and Square-attack) because the latter type has an additional access to local models.

---

> > > ### Author Response · Authors · 2022-08-08
> > > **Additional comparisons as requested**
> > >
> > > 1. Thank you for your suggestions. We feel that we have already done a comprehensive literature survey. As we mentioned in our previous response, we focused on methods that require a small number of queries (less than 500) and achieve a high fooling success rate (close to 90%). This rules out several papers that combine the transfer and (pure) query based attacks but are not query efficient.
> > >
> > > 2. We performed an experiment that combines [3*] and [4*] following your exact recommendations. We used the same (local) surrogate ensemble and images/labels that we used in our paper. For each image, we generate a perturbation using the surrogate ensemble and test transferability. If the attack fails to transfer, we deploy the Square-attack starting with the candidate perturbation. We use the official code from [4*]. Following the same setting in our Table 1 and Figure 2, we perform targeted attack on DenseNet121 with a perturbation budget of 16/255. The transfer rate of initial perturbed images is 75.5%. We attack the remaining 24.5% failed perturbed images using square attack by allowing a maximum query count of 500 (same setting as other baseline methods). On this subset of images, we observed a 33.9% fooling rate with query count (mean ± std): 238.2 ± 127.4. Overall, including the images that can initially transfer, the combination of [3*] and [4*] achieves a fooling rate of 83.8%, with query count (mean ± std): 24.5 ± 81.4. In comparison, our method achieves a 98.9% fooling rate with a query count of 1.7 ± 3.0. Similar trends appear for other victim models. Our main takeaway is that even though the surrogate ensemble provides highly transferable perturbation or perturbations that can be used as initialization for query-based optimization methods. The query-based methods lose their advantage by querying over a high dimensional image space. Our method searches over the weights of the ensemble loss, which is very low dimension and provides query efficiency.
> > >
> > > Table: Comparison of our method and [3*,4*] combination in terms of attack fooling rate and query count (mean ± std). Targeted attack settings are the same as used in Table 1 of our paper.
> > > | Victim model | Combination of [3*] and [4*] |        Ours       |
> > > |--------------|:----------------------------:|:-----------------:|
> > > |     VGG19    |      64.7% ; 34.1 ± 99.2     | 93.2% ; 2.8 ± 5.1 |
> > > | DenseNet121  |      83.8% ; 24.5 ± 81.4     | 98.9% ; 1.7 ± 3.0 |
> > > |   ResNext50  |      84.3% ; 24.7 ± 81.4     | 99.0% ; 1.8 ± 3.2 |

---

> > > > ### Comment · Reviewer_vphn · 2022-08-08
> > > > **My major concerns are well addressed**
> > > >
> > > > I appreciate the efforts from the authors and my raised concerns are well addressed.

---

### Author Response · Authors · 2022-08-02
**Summary response**

We thank all the reviewers for their careful and insightful comments. We very much appreciate  that reviewers recognized our work as a simple and effective strategy that leads to significant improvements.

The main contribution and novelty of this paper is that we create potent adversarial examples using a perturbation machine (PM) that is defined as a weighted loss over an ensemble of surrogate models. To fool a victim blackbox model, we generate perturbations by searching over the weights of the surrogate ensemble loss. Since the search dimension is the same as the number of models in the ensemble, our method requires a small number of queries. This is in contrast to most of the existing approaches that query over the image space and require a significantly larger number of queries. For instance, we demonstrate that our method provides more than a  90% success rate using an average of 3 queries per image in most cases (which is at least 30x fewer queries than that required by state-of-the-art methods).

The main concerns brought up by the reviewers are about comparison with baseline methods, differences in model architectures and training data between the surrogate models used for training and the targeted blackbox models.

We believe that the paper presents a comprehensive set of experiments (and comparisons with state-of-the-art methods) using a large variety of model architectures, different tasks (classification and object detection), and real-life blackbox models (Google Cloud VIsion API). Our framework is generic and can be used for other tasks (e.g. segmentation, pose estimation);

We present our detailed responses to all of the concerns in individual comments. We hope that our responses will address the reviewers’ concerns, and they will consider increasing their ratings. We look forward to engaging in further discussions, and answering any further questions that the reviewers may have for us.

---

### Author Response · Authors · 2022-08-06
**Request your attention**

Dear Reviewers and Chairs,

We hope you have received our response and had a chance to read them. We eagerly await any additional question or comment you may have on our paper.

Based on our understanding, main reviewer comments were about comparison and evaluation. We have provided detailed response to all the comments and provided additional evidence on why our method is superior to the current state-of-the-art. We hope our response has addressed the reviewer concerns. If we can provide any additional information, please let us know.

We also hope that you can review the ratings of the paper based on our response. We feel the current ratings do not match the contributions of this paper.

Thank you for your time and consideration.

---

### Meta-Review · Area_Chair_sRAd · 2022-08-27

**Recommendation:** Accept
**Confidence:** Certain

**Metareview:**

This paper proposes BASES, a query-efficient black-box adversarial attack by first generating adversarial perturbation with gradient-based attack using a weighted ensemble of surrogate models. The perturbed image is used to query the target model and its feedback is used to update the weights via zeroth-order optimization. The method is simple, intuitive and well-presented, and the authors show that it achieves a high attack success rate using a very limited query budget. The method can be used for both score-based and hard-label attacks, and experiment on Google Cloud Vision demonstrates real-world applicability.

The most common concern among reviewers is the method’s dependence on ensemble diversity. AC agrees that this is an inherent limitation as Figure 3 in the paper shows drastically reduced attack success rate when using a smaller set of surrogate models. However, reviewers wjHs and vphn argued that the paper’s contributions outweigh this limitation. AC agrees with this characterization and recommends acceptance, and encourages the authors to incorporate additional results from the rebuttal in the camera ready version.


**Award:**

No

---

### Decision · Program_Chairs · 2022-09-14

Accept